# Efficient and Transferable Agentic Knowledge Graph RAG via Reinforcement Learning

## Abstract

Knowledge-graph retrieval-augmented generation (KG-RAG) couples large language models (LLMs) with structured, verifiable knowledge graphs (KGs) to reduce hallucinations and expose reasoning traces. However, many KG-RAG systems compose multiple LLM modules (e.g planning, reasoning, and responding), inflating inference cost and binding behavior to a specific target KG. To address this, we introduce `KG-R1`, an agentic KG retrieval-augmented generation (KG-RAG) framework through reinforcement learning (RL). `KG-R1` utilizes a single agent that interacts with KGs as its environment, learning to retrieve at each step and incorporating the retrieved information into its reasoning and generation. The process is optimized through end-to-end RL. In controlled experiments across Knowledge-Graph Question Answering(KGQA) benchmarks, our method demonstrates both *efficiency* and *transferability*: Using Qwen-2.5-3B, `KG-R1` improves answer accuracy with fewer generation tokens than prior multi-module workflow methods that use larger foundation or fine-tuned models. Furthermore, `KG-R1` enables *plug and play*: after training, it maintains strong accuracy on new KGs without modification. These properties make `KG-R1` a promising KG-RAG framework for real-world deployment. Our code is publicly available at ⌂ `anonymous.4open.science/r/RL_KG-4B4E`.

## 1 Introduction

Retrieval-augmented generation (RAG) has gained popularity as a way to enhance large language models (LLMs) with access to external knowledge, thereby reducing hallucinations and improving accuracy in knowledge-intensive tasks. Recent research has extended this idea to knowledge graph retrieval-augmented generation (KG-RAG), where *knowledge graphs (KGs)* are leveraged as the retrieval source. A KG is a structured representation of knowledge in the form of entities (nodes) and their relationships (edges) that encodes factual knowledge in a graph format. Augmenting LLMs with KGs has proven effective not only in mitigating the knowledge bottleneck, but also in improving reasoning performance over complex multi-hop relations and enhancing adaptation to continually evolving real-world information (Sun et al., 2024; Luo et al., 2024; Wang et al., 2025c). These properties make KG-RAG especially promising in high-stakes domains, such as medical consultation and legal analysis (Xiong et al., 2023; Cui et al., 2024).

As shown in Figure 1, typical KG-RAG adopts a *modular workflow* consisting of four primary subtasks: *Retrieval* to query facts from KGs, *Reasoning* to process the retrieved infor-

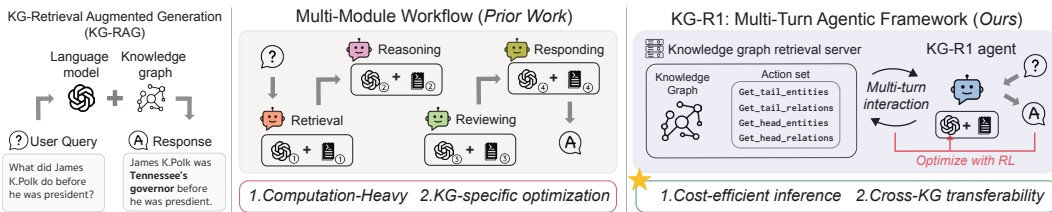

Figure 1: Overview of `KG-R1`, a multi-turn agentic framework for KG-RAG trained with reinforcement learning. The framework enables cost-efficient inference and demonstrates strong cross-KG transferability.

mation, *Reviewing* to verify logical consistency, and *Responding* to synthesize the final answer (Baek et al., 2023; Sun et al., 2024; Luo et al., 2024; Wang et al., 2025c). Each distinct subtask is handled by specialized LLM modules through two main methods: (i) prompt-based with task-specific instructions, often including in-context demonstrations (Baek et al., 2023); and (ii) fine-tuned modules tailored to particular tasks (e.g., SPARQL generation (D'Abramo et al., 2025; Lee & Shin, 2024) or relation extraction (Yao et al., 2019)) on specific KGs.

Despite improved reasoning accuracy, the real-world deployment of such workflows faces two key challenges, **high computational cost** and **lack of generalization to new or updated KGs**. First, prompt-based methods that repeatedly call large foundation LLMs accumulate inference across stages and drive up latency, token usage, and compute (e.g., ToG and ReKnoS; see left panel of Fig. 2). Second, these prompted or fine-tuned modules are typically tuned to a particular KG's domain and schema (entity types, relations, constraints), often via curated in-context examples or KG-specific fine-tuning. As a result, performance does not transfer reliably when the domain shifts, the schema changes, or the system is deployed on a new KG (e.g., RoG; see right panel of Fig. 2).

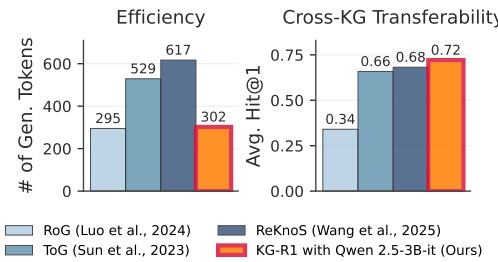

Figure 2: Prior multi-module methods are costly and do not transfer well across KGs. *Left:* mean end-to-end generated tokens per query on We-bQSP (Yih et al., 2016). *Right:* average Hit@1 over five out-of-training KGQA datasets (See Sec. 4.2). KG-R1 achieves *both* low token cost and strong cross-KG transferability.

To tackle these challenges, we introduce KG-R1, an agentic KG-RAG system that employs end-to-end multi-turn reinforcement learning (Jin et al., 2025; Zeng et al., 2025; DeepSeek-AI, 2025). As shown in Figure 1, the architecture of KG-R1 has two components: a single LLM agent and a KG retrieval server (as an environment). The KG retrieval server hosts the knowledge graph along with a set of retrieval actions. The LLM agent iteratively performs cycles of short reasoning followed by retrieval actions over multiple turns, with each decision informed by knowledge obtained from the KG retrieval server, and generates a final answer. Figure 2 demonstrates that KG-R1 achieves both high efficiency and strong cross-KG transferability *simultaneously* using a 3B-parameter model, outperforming prior methods. Our key contributions are summarized as follows:

**1. KG-R1 framework.** We introduce an agentic KG-RAG system (Section 3) that replaces multi-module pipelines with a *single* agent for KG-RAG, running against a lightweight KG server. The agent alternates between reasoning and retrieval actions over multiple turns, with the end-to-end trajectory optimized by RL using both *turn-wise* and *outcome-based* reward signals. Turn-wise rewards evaluate individual action effectiveness and adherence to formatting, while global rewards measure answer quality and retrieval relevance.

**2. Efficient inference.** By consolidating reasoning and retrieval into a *single-agent, near-continuous* workflow, KG-R1 achieves competitive reasoning accuracy with a small-parameter model while reducing token usage. This lowers latency and computational cost, making deployment feasible under tight budgets. Experiments demonstrate improved performance and efficiency compared to traditional multi-module workflows (Section 4.1).

**3. Plug-and-play transferability across diverse KGs.** KG-R1 easily transfers to diverse KGs and maintains strong KG-RAG performance (Section 4.2). The trained KG-R1 agent generalizes to new KGs without modification—backend KGs can be swapped without changing prompts or hyperparameters, and without fine-tuning. This enables zero-shot transfer to unseen knowledge graphs.

## 2 RELATED WORK

**KG-RAG.** Knowledge Graph Retrieval-Augmented Generation (KG-RAG) augments LLMs with structured knowledge graphs (KGs) to improve factuality and compositional reasoning. Early work grounds LLMs' generation by translating natural-language questions into executable graph queries

(e.g., SPARQL/Cypher), retrieving relevant subgraphs or answers, and feeding them back to the model (Ouyang et al., 2022; Izacard et al., 2023; Lee & Shin, 2024). More recent approaches adopt a modular LLM pipeline over KGs, interleaving natural-language reasoning with multi-stage planning, path search, and execution, where each stage uses a prompted or fine-tuned LLM (Luo et al., 2024; Sun et al., 2024; Wang et al., 2025c). Despite these advances, most systems rely on hand-engineered modules or prompt heuristics tied to a specific KG schema, which induce cost inefficiency and limit generalization. These challenges motivate our `KG-R1` framework: a single-agent KG-RAG approach that improves efficiency and transfers well to new KGs.

**Multi-turn RL for LLM.** Reinforcement learning (RL) has become central to equipping LLMs with step-by-step (chain-of-thought) reasoning behavior (OpenAI et al., 2024; DeepSeek-AI, 2025). RL-enhanced models yield substantial gains in math and coding (Le et al., 2022; Chervonyi et al., 2025), and broader complex reasoning tasks. More recently, RL has been applied to agentic LLMs that invoke external tools (e.g., bash terminals and APIs) or interact with knowledge bases, improving tool use (Qin et al., 2024) and retrieval-augmented generation (RAG) (Jin et al., 2025; Wang et al., 2025a) to facilitate the tool use or RAG. Building on these advances, our work, `KG-R1` adopts end-to-end RL as the core optimization algorithm for agentic KG-RAG framework.

**Distinction from RL-based Web Search Methods.** Recent works such as Search-R1 (Jin et al., 2025) and R1-Searcher (Song et al., 2025) apply RL to web-based RAG, where agents issue search queries over unstructured documents. While `KG-R1` shares the RL-optimized retrieval paradigm, it addresses KG-specific challenges: (1) structured graph traversal via 1-hop entity-relation operations rather than document retrieval; (2) schema-agnostic transfer across heterogeneous KGs (Freebase, Wikidata, temporal KGs) without retraining; and (3) multi-hop compositional reasoning requiring iterative graph exploration. These distinctions motivate a KG-specific RL formulation beyond web search methods.

## 3 KG-R1: AN AGENTIC KG-RAG FRAMEWORK

### 3.1 PROBLEM DEFINITION

KG–RAG spans many applications, including conversational assistants (Chaudhuri et al., 2021), recommendation systems (Wang et al., 2025b), and open-domain QA (Zhu et al., 2025). In this work, we instantiate and evaluate our approach on Knowledge Graph Question Answering (KGQA), which provides a grounded testbed for KG-RAG: ground-truth answers are tied to a fixed KG, evaluation is verifiable and intermediate graph reasoning is interpretable. We now formalize knowledge graphs and KGQA tasks.

**Knowledge graphs.** A knowledge graph (KG) is a graph-structured representation of real-world knowledge that encodes factual information as triples of entities and their relationships. We denote a KG as $G = \{ \langle e, r, e' \rangle \mid e, e' \in \mathcal{E}, \ r \in \mathcal{R} \}$, where $\mathcal{E}$ and $\mathcal{R}$ denote the sets of entities and relations, respectively, and each triple $\langle e, r, e' \rangle$ represents a directed edge from entity $e$ to entity $e'$ via relation $r$. For example, there is an edge `capital_of` from entity `Springfield` to entity `Illinois`.

**Knowledge Graph Question Answering (KGQA)** is a reasoning task based on KGs. Consider a dataset $D = \{(q, G, A_q)\}$, where $q$ is a natural language question, $G$ is a KG, and $A_q \subseteq \mathcal{E}$ is the ground-truth answer set paired with the question. For each gold answer $e^* \in A_q$, there exist one or more *reasoning paths* in $G$ that connect anchor entities mentioned in $q$ to $e^*$. A *reasoning path* is a sequence of labeled edges $r_1, \ldots, r_\ell \in \mathcal{R}$ instantiated by entities $e_0, \ldots, e_\ell$ such that $Z : e_0 \xrightarrow{r_1} e_1 \xrightarrow{r_2} \cdots \xrightarrow{r_\ell} e_\ell$, where $e_0$ is an anchor entity mentioned in $q$ and $e_\ell = e^* \in A_q$, and $\mathcal{Z}(q, G, A_q)$ denote the set of all valid reasoning paths for question $q$ over $G$ with respect to ground-truth answers $A_q$. In solving KGQA, given $q$ and $G$, a model attempts to discover a subset of valid reasoning paths from $\mathcal{Z}(q, G, A_q)$ and predicts $\hat{A}_q$ based on the terminal entities of the discovered paths. The model's performance is evaluated by comparing $\hat{A}_q$ with the ground-truth $A_q$. As an example, to answer the question "What is the capital of the U.S. state whose largest city is Chicago?", a valid reasoning path is

$$\text{Chicago} \xrightarrow{\texttt{located\_in\_state}} \text{Illinois} \xrightarrow{\texttt{capital}} \text{Springfield}$$

This path belongs to $\mathcal{Z}(q, G, A_q)$ and leads to the correct prediction $\hat{A}_q = \{\text{Springfield}\}$.

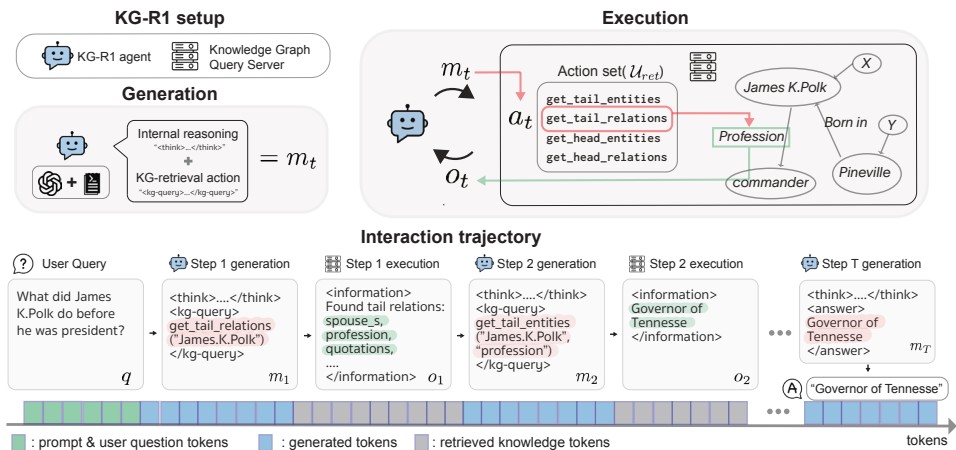

Figure 3: `KG-R1` framework: a single LLM agent undergoes multi-turn generation–execution loop with a schema-agnostic KG retrieval server and responds with the final answer.

## 3.2 KG-R1 FRAMEWORK

`KG-R1` casts KG-RAG as a multi-turn interaction with a KG interface (KG retrieval server). We prioritize two design principles. First, we design a *single-agent* architecture that simplifies deployment and enables efficient, low-cost inference. Second, we build a *schema-agnostic* KG retrieval server that avoids KG-specific assumptions and remains portable across varied KGs. Given a KGQA dataset $D = \{(q, G, A_q)\}$, we set up a KG retrieval server and a `KG-R1` agent.

**KG retrieval server.** The server hosts the knowledge graph $G$ and provides a set of retrieval actions $a \in \mathcal{U}_{\text{ret}}$ that enable graph traversal through 1-hop operations:

$$\mathcal{A}_{\text{ret}} = \big\{ \texttt{get\_tail\_relations}, \texttt{get\_head\_relations}, \texttt{get\_tail\_entities}, \texttt{get\_head\_entities} \big\}$$

$$\texttt{get\_tail\_relations}(e) := \{\, r \in \mathcal{R} \mid \exists e' \in \mathcal{E} : (e, r, e') \in G \,\},$$
$$\texttt{get\_head\_relations}(e') := \{\, r \in \mathcal{R} \mid \exists e \in \mathcal{E} : (e, r, e') \in G \,\},$$
$$\texttt{get\_tail\_entities}(e, r) := \{\, e' \in \mathcal{E} \mid (e, r, e') \in G \,\},$$
$$\texttt{get\_head\_entities}(r, e') := \{\, e \in \mathcal{E} \mid (e, r, e') \in G \,\}.$$

The retrieval action set $\mathcal{U}_{\text{ret}}$ is sufficient for *realizing* any reasoning path $Z : e_0 \xrightarrow{r_1} e_1 \xrightarrow{r_2} \cdots \xrightarrow{r_\ell} e_\ell$ as a finite action sequence that arrives at $e_\ell \in A_q$ (i.e., the terminal entity is in the answer set). Forward traversal along $Z$ is implemented by $\texttt{get\_tail\_entities}(e_{i-1}, r_i)$ for $i = 1, \ldots, \ell$ and backward traversal is implemented by $\texttt{get\_head\_entities}(r_i, e_i)$. Notably, this choice of the retrieval action set *guarantees* completeness and schema-free transferability, as formalized by propositions 3.1 and 3.2, whose proofs are provided in Appendix A.1.

**Proposition 3.1** (**Retrieval Action Set Completeness**). *For any reasoning path $Z : e_0 \xrightarrow{r_1} \cdots \xrightarrow{r_\ell} e_\ell$ in $G$, there exists an action sequence in $\mathcal{U}_{\text{ret}}$ of length at most $\ell+1$ whose output includes $e_\ell$.*

**Proposition 3.2** (**Schema-Free Transferability**). *The semantics of $\mathcal{U}_{\text{ret}}$ depend only on directed triples $(e, r, e')$, so any fixed client policy transfers across heterogeneous directed KGs by replacing $G$ without redefining $\mathcal{U}_{\text{ret}}$.*

**KG-R1 Agent.** We model a single *LLM agent* that interacts with a KG retrieval server in a multi-turn setting. The agent undergoes initialization followed by a loop of generation and execution (see Figure 3). At initialization, a lightweight base LLM is configured with an instruction prompt $p$ (see the box below for the prompt template) containing general reasoning instructions, the user question $q$, and the KG retrieval server instructions (Table D.2). At each turn $t \leq H$, where $H$ is the maximum turn limit, the agent first undergoes the **generation** phase where it produces a response $m_t$ comprising two parts: (1) an internal reasoning wrapped in `<think>...</think>`, and (2) an action, which is either a KG retrieval action wrapped in `<kg-query>...</kg-query>`, or a final answer wrapped in `<answer>...</answer>`.

---

**Prompt template for KG-R1**

You are a helpful assistant. Answer the given question. You can query from knowledge base provided to you to answer the question. You can query knowledge up to [H] times. You must first conduct reasoning inside <think>...</think>. If you need to query knowledge, you can set a query statement between <kg-query>...</kg-query> to query from knowledge base after <think>...</think>. When you have the final answer, you can output the answer inside <answer>...</answer>.
KG Query Server Instruction : [KG_query_server_instruction]
Question: [question].
Assistant:

---

In the following **execution** phase, we parse $m_t$ into an *action* $a_t \in \mathcal{U}_{\text{ret}} \cup \{\text{Answer}\}$ using an exact-match parser $\Psi$ (i.e., $a_t = \Psi(m_t)$). If $a_t \in \mathcal{U}_{\text{ret}}$, executing it yields an *observation* $o_{t+1}$ (i.e., retrieved entities, relations, or an error message in case the action does not generate a valid retrieval), which is appended to the dialogue context prior to the next turn. If $a_t = \text{Answer}$, the interaction terminates: the content inside <answer>...</answer> is extracted and post-processed (normalization, deduplication, and entity resolution) to produce the predicted answer set $\hat{A}_q$. Given interaction trajectory $\tau = (p, (m_1, o_1), \dots, (m_t, o_t))$ with $t \leq H$, the KG-R1 agent's *policy* $\pi_\theta(m_t \mid \text{context}_t)$ is defined over token sequences $m_t$ and governs how textual responses are generated.

### 3.3 KG-R1 TRAINING WITH RL

Our goal is to find the KG-R1 agent's policy $\pi_\theta(m_t \mid \text{context}_t)$ that effectively retreive reasoning paths on $G$ via multi-turn interaction and generate the accurate answer $\hat{A}_q = A_q$. To this end, we optimize the base LLM with reinforcement learning, using a GRPO-style objective (DeepSeek-AI, 2025) in a multi-turn setting (Qian et al., 2025; Jin et al., 2025; Zeng et al., 2025). Our reward function combines action-level and outcome-based signals. The overview of the training procedure refers to Algorithm 1 (Appendix).

**Reward Objective.** To effectively train the KG-R1 agent, we combine verifiable *turn* rewards with outcome-level *global* rewards. **Turn rewards** ($r_t^{\text{turn}}$) provide local signals at each step as the sum of three components: (i) *format validity* $v_{\text{fmt}}(m_t)$ checks that $m_t$ contains both reasoning and a well-formed action $a_t \in \mathcal{A}$; (ii) *KG query* $v_{\text{kg}}(a_t, o_{t+1})$ checks that executing $a_t$ yields meaningful, schema-valid retrieval in $o_{t+1}$; and (iii) *answer* $v_{\text{ans}}(m_T)$ checks the final answer's format/consistency on the final turn. the turn rewards are computed as follows with weight $w_{fmt}, w_{kg}, w_{ans}$:

$$r_t^{\text{turn}} = w_{\text{fmt}}\, v_{\text{fmt}}(m_t) + w_{\text{kg}}\, v_{\text{kg}}(a_t, o_{t+1}) + w_{\text{ans}}\, v_{\text{ans}}(m_T). \tag{1}$$

**Global rewards** summarize trajectory-level outcomes as the sum of the following: (i) a final-answer *F1* score over the predicted set $\hat{A}_q$ vs. ground-truth answer set $A_q$; and (ii) a *retrieval score* $v_{ret}$ that is 1 if any gold entity appears anywhere in the retrieved information along the executed reasoning path, and 0 otherwise.

$$R^{\text{global}} = w_{\text{F1}} \cdot \text{F1}(\hat{A}_q, A_q) + w_{\text{ret}} \cdot v_{ret} \tag{2}$$

**Group-relative turn-level credit assignment and optimization.** To convert rewards into token-level credit, we use a group-relative, turn-level credit assignment inspired by Zeng et al. (2025). We collect $N$ rollouts per query $q$. For each episode $\tau^{(n)}$, we compute turn rewards $r_t^{\text{turn},(n)}$ and a global reward $R^{\text{global},(n)}$, and then form turn-specific returns with $\lambda$ as the normalization factor:

$$G_t^{(n)} = r_t^{\text{turn},(n)} + \lambda\, R^{\text{global},(n)}. \tag{3}$$

Let $\mathcal{S} = \{(n, t) : t \leq T^{(n)}\}$ be the set of all turns across the $N$ rollouts. The turn-level advantage $A_t^n$ is calculated using a *single* group baseline $\bar{G}$ that averages over $\mathcal{S}$:

$$A_t^{(n)} = \frac{G_t^{(n)} - \bar{G}}{\sigma_G + \varepsilon} \quad \text{where} \quad \bar{G} = \frac{1}{|\mathcal{S}|} \sum_{(n,t) \in \mathcal{S}} G_t^{(n)}, \ \sigma_G = \sqrt{\frac{1}{|\mathcal{S}|} \sum_{(n,t) \in \mathcal{S}} \left(G_t^{(n)} - \bar{G}\right)^2}.$$

where $\varepsilon$ is a small constant for numerical stability.

**RL update.** Using turn-level credit $A_t^{(n)}$, we optimize the agent's policy $\pi_\theta$ with a GRPO-style objective $\mathcal{J}$ :

$$\mathcal{J}(\theta) = \mathbb{E}\left[ \sum_{n,t,i} \min\left( \rho_{t,i}^{(n)}\, \widetilde{A}_{t,i}^{(n)},\, \mathrm{clip}\left( \rho_{t,i}^{(n)},\, 1-\epsilon,\, 1+\epsilon \right) \widetilde{A}_{t,i}^{(n)} \right) - \beta\, \mathrm{KL}\left( \pi_\theta(\cdot \mid h_{t,i}^{(n)}) \,\|\, \pi_0(\cdot \mid h_{t,i}^{(n)}) \right) \right]$$

with $\widetilde{A}_{t,i}^{(n)} = m_{t,i}^{(n)} A_t^{(n)}$. $\pi_\theta$ is the current policy, and $\pi_{\theta_{\text{old}}}$ is the behavior policy used for sampling; $\pi_0$ is a fixed reference policy used only for KL regularization (weight $\beta$). $\rho_{t,i}^{(n)} = \exp\big( \log \pi_\theta(y_{t,i}^{(n)} \mid h_{t,i}^{(n)}) - \log \pi_{\theta_{\text{old}}}(y_{t,i}^{(n)} \mid h_{t,i}^{(n)}) \big)$ is the importance ratio that corrects off-policy sampling at token $i$. We maximize $\mathcal{J}(\theta)$ by gradient ascent. The proposed group-based credit assignment per turn produces stabilized and effective signals in the multi-turn interaction decisions. We provide comprehensive ablation studies of each component of our reward design in Section 4.3.

## 3.4 CROSS-KG TRANSFER: PLUG-AND-PLAY

We propose a plug-and-play approach for `KG-R1` that enables KG–RAG to operate across different knowledge graphs without retraining. Let $\pi_{\theta|\mathcal{D}}$ denote the policy of the `KG-R1` agent trained on a KGQA dataset $\mathcal{D} = \{(q, G, A_q)\}$. For a new KGQA dataset $\mathcal{D}^* = \{(q^*, G^*, A_{q^*})\}$, we replace the backend KG of the retrieval server with $G^*$ (*plug*) and evaluate $\pi_{\theta|\mathcal{D}}$ on $\mathcal{D}^*$ without any modification (*play*). This approach requires no task-specific fine-tuning, data relabeling, or architectural changes.

This plug-and-play capability stems from two design choices in `KG-R1`. First, the KG server exposes a schema-agnostic action set consisting solely of 1-hop retrieval operations, which are universally compatible with directed KGs. This contrasts with SPARQL-generation approaches in prior work that depend on KG-specific syntax and schema knowledge. Second, the agent's strategy for retrieving and using KG evidence is learned in a way that is independent of any particular schema, enabling immediate transfer to new domains.

## 4 EXPERIMENTS

We assess `KG-R1` along two axes. (1) *Performance and Efficiency on Trained KGs:* We train `KG-R1` on the training splits of KGQA benchmarks and measure answer quality and inference cost on their held-out test splits. (2) *Cross-KG transferability*: We test how well `KG-R1` carries over to KGs out of training. In both parts, we compare against established baselines to support the `KG-R1` results.

### 4.1 PERFORMANCE AND EFFICIENCY ON TRAINED KGS

**Models.** We use Qwen2.5-3B-it (Qwen et al., 2025) as our base model. For the other baseline methods, we use GPT-4o-mini, GPT-3.5, and Llama2-7B-it, following the setups used in prior work.

**Datasets.** We mainly evaluate `KG-R1` in-domain on (1) WEBQSP (Yih et al., 2016)—natural-language questions over Freebase, mostly 1–2 hop, using the official 3,098/1,639 train/test QA split; and (2) COMPLEXWEBQUESTIONS (CWQ) (Talmor & Berant, 2018)—compositional, multi-hop questions over Freebase with a 24,649/3,531 train/test QA split. For scalability, we extract 2-hop subgraphs for each question as in RoG (Luo et al., 2024). See Appendix B for detailed sources.

**Metrics.** Following prior work (Luo et al., 2024; Sun et al., 2024) to evaluate KGQA performance, we use *F1 score* to consider multiple valid predictions over answer sets and *Hit@1* (1 if the gold answer appears in the single search, 0 otherwise). For efficiency, we report *generated tokens* (completion-only), measured end-to-end per query, aggregated across all turns $(1 \dots H)$. This serves as a proxy for inference time and compute cost. For fair comparison, all token counts are computed with the Qwen2.5 tokenizer. We also report the *number of modules* to compare workflow complexity, and we analyze single-query latency and batched throughput on a single GPU node.

**Baselines.** We compare three classes of approaches: (1) *LLM-only* methods that do not access an external KG—Vanilla and Chain-of-Thought (CoT) prompting; (2) *prior KG-augmented* pipelines: RoG (Luo et al., 2024), which uses a *fine-tuned* modular workflow with LLaMA2-7B-it as the

backbone, and ToG (Sun et al., 2024) and REKNOS (Wang et al., 2025c), which use a *prompt-based* modular workflow with GPT-3.5; and (3) our method `KG-R1`. For LLM-only baselines, we employ LLM-as-Judge (Zheng et al., 2023) using gpt-4o-mini to semantically match predicted answers to ground truth. Full details for baselines are provided in Appendix C.

$N$**-response sampling with `KG-R1`.** Prior works often incorporated $k$-beam search (Lan & Jiang, 2020; Sun et al., 2024; Luo et al., 2024), retrieving $k$ reasoning paths for enhanced search range on KGs and reporting improved performance. Following this idea, we propose `KG-R1` $N$-response sampling: we run $N$ independent response samplings per question and collect predicted answer sets $\{\hat{A}_q^{(i)}\}_{i=1}^N$. The KG-R1 agent, using a nonzero decoding temperature, yields diverse answers in each run due to its stochastic policy. We then take the union $\hat{A}_{q,\text{N-runs}} = \bigcup_{i=1}^N \hat{A}_q^{(i)}$, and compute F1 and *Hit@1* on this union. This $N$-response sampling broadens search without extra orchestration, and cost scales roughly linearly with $N$ since runs can be executed in parallel.

**Implementation details.** We use VERL (Sheng et al., 2024) and vLLM-backed decoding (Kwon et al., 2023) for implementing the `KG-R1` agent. Training uses distributed optimization with gradient checkpointing and offloading. Unless stated, we set $H = 5$ turns and $N=16$ rollouts per query. Additional implementation details (e.g., hyper-parameters) are provided in Appendix D.4.

### 4.1.1 KG-R1 TRAINING STABILITY AND EFFECTIVENESS

Figure 4 shows steady F1 improvement that plateaus after convergence during `KG-R1` training on WebQSP and CWQ. The validation F1 scores track the training curve, indicating the agent learns generalization. Results are reproducible across three random seeds (Figure 6, Appendix), showing strong convergence with low variance. Overall, `KG-R1` training yields stable, reproducible gains in end-to-end KG-RAG performance.

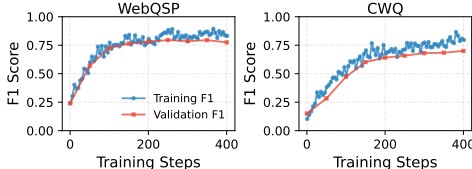

Figure 4: F1 score over `KG-R1` training on WebQSP and CWQ for Qwen2.5-3B-it. Training (blue) and validation (red).

### 4.1.2 INFERENCE ACCURACY AND EFFICIENCY RESULTS

**A1. `KG-R1` demonstrates strong accuracy on trained KGs.** Table 1 reports accuracy and efficiency. On both WebQSP and CWQ, a single run ($N=1$) of `KG-R1` achieves competitive F1 and Hit@1 compared with non-KG and prior KG methods, including those using larger foundation models. With *$N$-response sampling ($N=3$)*, `KG-R1` surpasses prior LLM+KG systems that also use $k$-beam search by a clear margin (WEBQSP: F1 77.5→85.8, Hit@1 84.7→91.7; CWQ: F1 70.9→81.0, Hit@1 73.8→83.9). Overall, these results show that a lightweight `KG-R1` agent delivers strong KG-RAG performance relative to previous baselines.

**A2. `KG-R1` achieves efficient inference.** Table 1 reports efficiency (generation tokens and number of modules). For a single run ($N=1$), `KG-R1` produces ∼60 generation tokens per turn (300–370 tokens at $H=5$), substantially below prior prompt-based LLM+KG systems (e.g., ToG ∼520 and ReKnoS ∼600–650) and comparable to the fine-tuned method RoG (∼250–300). This small generation-token budget, coupled with a lightweight model, implies improved throughput and latency. Under $N=1$, the *total tokens* are comparable to prior methods, indicating a similar memory (KV-cache) footprint. With *$N$-response sampling ($N=3$)*, token usage scales nearly linearly, exposing a tunable accuracy–efficiency trade-off via $N$.

**A3. Latency and throughput analysis.** On a single NVIDIA H100 GPU, we measure (i) single-query end-to-end latency and (ii) batched throughput. The single-query latency averages $6.4 \pm 1.5$ s per question. Batched throughput reaches 3.7 samples per second at batch size 64. The results suggest the feasibility of KG-R1's single-node deployment. See Appendix E.2 for the full results.

**A4. `KG-R1` outperforms Search-R1 in both accuracy and efficiency.** We compare `KG-R1` against Search-R1 (Jin et al., 2025), an RL-based retrieval-augmented reasoning method using embedding-based search, adapted to KGQA by converting KG triples to text with BGE embeddings (see Appendix G for details). Search-R1 fails to learn stable KGQA policies: despite extensive tuning, it achieves only 58 F1 on WebQSP (peak during training) before collapsing to near-zero within ∼50

Table 1: Performance and efficiency comparison of KGQA methods on WebQSP and CWQ datasets. The max turn number set to $H = 5$. All scores are percentages.

| Method | Model | Modules | WebQSP | | | | CWQ | | | |
| | | | Performance | | Efficiency | | Performance | | Efficiency | |
| | | | F1 | Hit@1 | Total | Gen | F1 | Hit@1 | Total | Gen |
| --- | --- | --- | --- | --- | --- | --- | --- | --- | --- | --- |
| *LLM-only Methods* | | | | | | | | | | |
| Vanilla | Qwen2.5-3B-it | 1 | 29.4 | 46.6 | 95 | 30 | 16.6 | 21.1 | 104 | 42 |
| Vanilla | Qwen2.5-7B-it | 1 | 33.2 | 50.9 | 95 | 67 | 20.7 | 25.7 | 104 | 92 |
| Vanilla | LLaMA2-7B-it | 1 | 37.4 | 54.5 | 114 | 255 | 20.7 | 24.8 | 123 | 255 |
| COT | Qwen2.5-3B-it | 1 | 30.6 | 47.6 | 131 | 192 | 17.3 | 21.4 | 140 | 216 |
| COT | Qwen2.5-7B-it | 1 | 35.3 | 53.5 | 131 | 194 | 22.5 | 27.1 | 140 | 212 |
| COT | LLaMA2-7B-it | 1 | 33.8 | 51.6 | 165 | 255 | 19.0 | 23.1 | 174 | 255 |
| *LLM+KG Methods* | | | | | | | | | | |
| RoG | LLaMA2-7B-it | 2 | 70.8 | **85.7** | 1.2K | 295 | 56.2 | 62.6 | 1.1K | 266 |
| ToG | GPT-3.5 | 4 | 72.8 | 76.2 | 3.5K | 529 | 52.9 | 57.1 | 3.7K | 520 |
| ToG 2.0 | GPT-3.5 | 5 | 74.5 | 77.8 | 39.K | 605 | 65.8 | 68.9 | 3.8K | 650 |
| ReKnoS | GPT-4o-mini | 3 | 73.7 | 81.1 | 3.1K | 617 | 64.7 | 66.8 | 4.1K | 752 |
| GCR | Qwen2.5-3B-it | 2 | 65.5 | 72.3 | 421 | 59 | 46.3 | 56.0 | 453 | 58 |
| GNN-RAG | Qwen2.5-3B-it | 2 | 72.0 | 81.2 | 480 | 31 | 59.4 | 61.7 | 818 | 16 |
| PoG | Qwen2.5-3B-it | 3 | 7.1 | 19.4 | 2.3K | 248 | – | – | 2.3K | 284 |
| *KG-R1 (Our Methods)* | | | | | | | | | | |
| **KG-R1 (1 run)** | Qwen2.5-3B-it | 1 | **77.5** | 84.7 | 3.2K | 302 | **70.9** | **73.8** | 3.3K | 377 |
| **KG-R1 (3 runs)** | Qwen2.5-3B-it | 1 | 85.8 | 91.7 | 9.7K | 906 | 81.0 | 83.9 | 10.0K | 1.1K |

steps, compared to `KG-R1`'s stable 77.5 F1. On CWQ, Search-R1 peaks at 21 F1 versus `KG-R1`'s 70.9 F1. Moreover, `KG-R1` is approximately $5\times$ faster (216ms vs 1116ms per question) because embedding-based retrieval incurs $\sim$100–150ms overhead per query, while direct graph lookups require only $\sim$2–5ms.

## 4.2 CROSS-KG TRANSFER VIA PLUG-AND-PLAY

In this subsection, we evaluate the plug-and-play approach (Sec. 3.4) for zero-shot generalization of our `KG-R1` agent on knowledge graphs outside its training distribution.

**Datasets** We evaluate the plug-and-play approach on diverse KGQA benchmarks spanning three categories of KGs. (1) *Freebase* type: SimpleQA (Bordes et al., 2015) and GrailQA (Gu et al., 2021), which share the Freebase KG but differ in question complexity and reasoning path distributions. (2) *Wikidata* type: T-REx (Elsahar et al., 2018) and QALD-10en (Usbeck et al., 2023) for cross-schema generalization. (3) *Temporal reasoning benchmark*: MultiTQ (Chen et al., 2023), which requires reasoning over the ICEWS temporal KG (Boschee et al., 2015) at day, month, and year granularities. Appendix B provides a summary of dataset sources, schema, and evaluation splits.

**Baselines** For *LLM-only methods*, we evaluate (i) Vanilla and (ii) Chain-of-Thought (CoT) prompting with Qwen2.5–3B-it. For *LLM+KG methods*, we report results for RoG, ToG, and ReKnoS without modifying their modules (i.e prompts and models). In addition, for each KGQA benchmark, we include one or two recent state-of-the-art *KG-specific methods* (trained/tuned on the target dataset with task-specific pipelines) and use their published test-set scores. Full baseline details are provided in Appendix C.3.

**A5. `KG-R1` transfers across diverse types of KGs.** Table 2 reports zero-shot, *plug-and-play* results. For comparability, we report **AVG.**, the F1/Hits@1 averaged across the five KGQA datasets, weighted by QA counts. `KG-R1` (1 run) substantially outperforms LLM-only baselines using the same base model—Vanilla (19.4/19.8 F1/Hits@1) and CoT (18.0/18.2)—reaching 64.0/68.3 when trained on WebQSP and 67.2/72.1 when trained on CWQ. The gains hold not only on Freebase-based QA but also on Wikidata and Temporal benchmarks, indicating that reinforcement-learned KG-RAG transfer across diverse KGs rather than narrow, in-domain improvements.

**A6. `KG-R1` achieves inference comparable to LLM+KG baselines.** Against LLM+KG baselines, KG-R1 (N=1) is slightly better on average: ToG (66.2/65.9) and ReKnoS (68.2/68.2). Notably, `KG-R1`, with a 3B model, delivers higher Hits@1 than both ToG and ReKnoS that uses

Table 2: Zero-shot cross-KG transferability of KG-R1. Agents are trained on WebQSP or CWQ and evaluated on new benchmarks by *swapping only* the KG-backend server (no policy retraining). **AVG.** is averaged across QA samples. The max turn number set to $H = 5$. All scores are percentages.

| Method | Trained KG | Freebase-based | | Wikidata-based | | Temporal | **AVG.** |
|---|---|---|---|---|---|---|---|
| | | SimpleQA | GrailQA | T-REx | QALD-10en | MultiTQ | |
| | | F1 / Hit@1 | F1 / Hit@1 | F1 / Hit@1 | F1 / Hit@1 | F1 / Hit@1 | F1 / Hit@1 |
| *LLM-only Methods* | | | | | | | |
| Vanilla Qwen 2.5 3B-IT | | 13.7 / 13.7 | 15.9 / 15.9 | 24.4 / 24.4 | 23.8 / 23.8 | 2.2 / 5.4 | 19.4 / 19.8 |
| CoT Qwen 2.5 3B-IT | | 10.9 / 10.9 | 18.2 / 18.2 | 22.0 / 22.0 | 25.9 / 25.9 | 1.9 / 3.9 | 18.0 / 18.2 |
| *LLM+KG Methods* | | | | | | | |
| KG-specific Baselines | | $79.6^1/80.6^1$ | $84.4^3/79.1^3$ | — / $85.1^5$ | $49.8^6/50.8^6$ | — / $79.7^8$ | |
| | | $76.1^2/77.1^2$ | $64.9^4/59.6^4$ | | $62.2^7/63.2^7$ | — / $38.0^9$ | |
| RoG | | 13.5 / 43.5 | 6.4 / 23.7 | 8.1 / 36.4 | 11.4 / 44.3 | 3.6 / 8.7 | 8.1 / 34 |
| ToG | | 50.2 / 53.6 | 63.9 / 68.7 | 79.3 / 76.4 | 48.7 / 50.2 | 25.1 / 27.9 | 66.2 / 65.9 |
| ToG 2.0 | | – | – | 80.7 / 77.5 | 51.9 / 54.6 | – | – |
| ReKnoS | | 50.7 / 52.9 | 65.8 / 71.9 | 81.4 / 78.9 | 50.6 / 53.8 | 28.5 / 31.1 | 68.2 / 68.2 |
| *KG-R1 (Our Methods)* | | | | | | | |
| **KG-R1 (1 run)** | WebQSP | 59.1 / 59.1 | 32.8 / 38.5 | 80.5 / 84.5 | 51.9 / 53.4 | 21.6 / 31.4 | 64.0 / 68.3 |
| **KG-R1 (1 run)** | CWQ | 64.6 / 64.7 | 42.8 / 50.2 | 81.3 / 85.6 | 55.9 / 57.7 | 27.1 / 38.9 | 67.2 / 72.1 |
| **KG-R1 (3 runs)** | WebQSP | 69.1 / 69.1 | 44.6 / 52.1 | 84.6 / 88.8 | 61.3 / 62.8 | 33.5 / 45.7 | 70.9 / 75.8 |
| **KG-R1 (3 runs)** | CWQ | 73.1 / 73.1 | 52.8 / 61.0 | 86.8 / 91.5 | 63.9 / 65.5 | 36.2 / 48.4 | 74.1 / 79.4 |

[1]SPARKLE [2]GETT-QA [3]SG-KBQA [4]ARG-KBQA [5]ATLAS [6]COT-SPARQL [7]DFSL-MQ [8]Prog-TQA [9]ARI ; see Table 7 for references. ToG 2.0 requires a Wikipedia-based KG with associated documents, hence only applicable to T-REx and QALD-10en.

strong foundation models. Increasing to N=3 runs boosts performance to 74.1/70.4 (CWQ-trained), making KG-R1 comparable to KG-specific baselines on average. We observe that RoG collapses (0/0 average) revealing brittle, schema-specific planning with fine-tuned model that does not transfer. Overall, KG-R1 attains strong transferability among plug-and-play methods with a lightweight model, supporting its practicality for real-world KG-RAG deployment across diverse KGs.

**A7. Training data affects transferability.** KG-R1 models trained on CWQ consistently outperform WebQSP-trained variants across all transfer targets, averaging 1–5% higher F1 and Hits@1. Given that CWQ provides a larger training set with greater multi-hop complexity than WebQSP, this suggests that exposure to more complex and diverse reasoning patterns yields superior generalization.

**A8. MedQA reveals KG-RAG limitations.** To understand transferability limitations, we evaluate on MedQA (Jin et al., 2021), a medical QA benchmark requiring clinical reasoning, using Drug-Bank Disease Database (DDB) and PrimeKG (Chandak et al., 2023). KG-R1 underperforms vanilla baselines (25.6% vs 31.6% accuracy with DDB), revealing a failure mode: medical KGs encode surface-level entity relationships rather than causal clinical reasoning paths, entity linking fails for 23% of medical terms due to terminology mismatch, and the agent cannot synthesize fragmented facts into coherent clinical decisions. This highlights that KG-RAG benefits depend critically on KG quality and domain alignment. Full analysis is provided in Appendix F.

## 4.3 ABLATION STUDIES

We conduct ablation studies for the three core components of KG-R1: reward design (Section 3.3), RL algorithm (Section 3.3), and KG-server implementation (Section 3.2) and base LLM size. We train Qwen-2.5-3B-it on WEBQSP and CWQ with maximum turn number $H = 5$ and report final step performance(F1/Hit@1/Ret.rate in Table 3). The training curves are shown in Figures 8, 9.

**A9. Reward function.** First, removing turn-level rewards yields the largest drop on both WebQSP (13.1% / 9.9% in F1 score and Hit@1, respectively) and CWQ (22.3% / 24.1%), highlighting the importance of per-turn validity/effectiveness signals. Next, disabling the turn-wise advantage calculation produces a substantial decline on WebQSP (35.8% / 27.4%) but only a moderate drop on CWQ (4.9% / 7.5%), indicating that turn-specific learning signals matter more for certain datasets. Finally, removing the retrieval reward significantly impacts WebQSP (40.7% / 31.5%) but marginally affects CWQ (1.2% / 3.9%), though it substantially reduces the retrieval success rate of gold entities (Ret.Rate) on both datasets, suggesting that it is essential for motivating exploration of the KG.

**A10. RL algorithm (GRPO vs. PPO).** Replacing GRPO with vanilla PPO (Ouyang et al., 2022) results in learning collapse (0.0 across all metrics). Under PPO, we observe *reward hacking* (see

Table 3: Ablation studies of `KG-R1` components on CWQ and WebQSP datasets.

| Method | WebQSP | | | CWQ | | |
|---|---|---|---|---|---|---|
| | F1 | Hit@1 | Ret.Rate | F1 | Hit@1 | Ret.Rate |
| *KG-R1 Default* | | | | | | |
| **KG-R1** | 77.2 | 84.3 | 76.8 | 66.8 | 69.7 | 51.4 |
| *Reward Ablation* | | | | | | |
| w/o Turn Rewards | 67.1 (-13.1%) | 76.0 (-9.8%) | 68.1 | 51.9 (-22.3%) | 52.9 (-24.1%) | 44.7 |
| w/o Turnwise Advantage | 49.6 (-35.8%) | 61.2 (-27.4%) | 22.8 | 63.5 (-4.9%) | 64.5 (-7.5%) | 49.4 |
| w/o Retrieval Reward | 45.8 (-40.6%) | 57.7 (-31.5%) | 18.6 | 66.0 (-1.2%) | 67.0 (-3.9%) | 47.2 |
| *Other Ablations* | | | | | | |
| w PPO | 0.0[†] (-100.0%) | 0.0 (-100.0%) | 0.0 | 0.0[†] (-100.0%) | 0.0 (-100.0%) | 0.0 |
| w/o Hierarchical Rel. Retr. | 76.8 (-0.6%) | 83.1 (-1.5%) | 76.0 | 55.5 (-16.9%) | 58.6 (-15.9%) | 39.7 |
| w Qwen-2.5-7B-it | 79.6[‡] (+3.1%) | 86.6 (+2.7%) | 83.4 (+8.6%) | 65.6[‡] (-1.8%) | 68.7 (-1.4%) | 58.8 (+14.4%) |

[†] PPO training crashed.     [‡] Peak values before 7B model training collapse.

examples in Appendix E.3.3): the agent fabricates "retrieved" content matching expected formats to earn reward. Since PPO's value critic (another LLM) cannot distinguish genuine KG retrievals from hallucinations, its advantage estimates become unstable, driving training collapse. This underscores the importance of relative advantage methods like GRPO for multi-step reasoning.

**A11. Retrieval format.** In our standard `KG-R1` setup, one-hop relations retrieved by `get_head_relations` and `get_tail_relations` are appended to the agent's context as a flat, comma-separated list (e.g., "rel1, rel2, ..."). This can lead to excessive token consumption when many relations are retrieved. Because Freebase relations use a compositional format (e.g., "domain.type.property"), we tested a hierarchical relation retrieval that groups relations into a tree to reduce redundancy (see Appendix E.3.2). Our ablation studies show that switching from flat lists to a hierarchical format reduces performance—WebQSP shows 0.5% drop, while CWQ experiences 16.9% drop. These results suggest that despite its higher token usage, the flat retrieval format provides more direct access to relation information, which proves critical for training.

**A12. Model scale.** The 7B model improves more quickly than the 3B model but exhibits training collapse (around 350 steps on WebQSP and 200 steps on CWQ). This aligns with reports that fast distribution shift during GRPO—amplified by larger models and higher update rates—drives entropy collapse and model divergence (Jin et al., 2025; Liu et al., 2025).

## 5 CONCLUSION

We propose `KG-R1`, a single-agent, multi-turn KG-RAG framework in which one LLM queries a lightweight KG server and is optimized via RL. Across KG-augmented QA benchmarks, `KG-R1` attains strong accuracy while using markedly fewer tokens and lower inference cost than prior work. It also supports *plug-and-play* transfer, retaining robust performance across KGs. Together, these results position `KG-R1` as a practical and deployable KG-RAG system for real-world use.

### 5.1 LIMITATIONS AND FUTURE WORK

**Restricted Expressivity of 1-hop Retrieval Actions.** `KG-R1` currently operates on 1-hop relational transitions. While this simplifies the policy and enables a KG-schema agnostic design, it may limit the agent's ability to efficiently retrieve high-level logical patterns or reason over long-range dependencies. Future work can incorporate more elaborate actions, such as multi-hop path retrieval or hierarchical retrieval, to improve retrieval efficiency.

**Dependency on Entity Linking.** The retrieval actions in `KG-R1` currently rely on exact string matching for entity names. Consequently, if entity matching fails (e.g., the agent generates semantically similar but textually distinct entities), the agent receives an error message and loses the reasoning context. Future work can address this issue by incorporating semantic-based entity linking (Wu et al., 2020) to allow for fuzzy matching and synonym resolution.

**Sparse Reward in Multi-Turn RL.** While our proposed multi-turn RL setup is effective, there is room for advancement in optimal training and policy learning. Future work could incorporate early-exit mechanisms (Dai et al., 2025) to reduce computational redundancy, as well as sequence-level policy optimization (Zheng et al., 2025) to mitigate the high variance inherent in token-level updates, thereby further enhancing RL training stability and performance.

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

APPENDIX

# A THEORETICAL PROOFS

## A.1 COMPLETENESS AND TRANSFERABILITY OF THE KG-SERVER

Here we provide robust proofs for completeness and transferability of $\mathcal{U}_{ret}$ in our KG retrieval server.

**Preliminaries.** Let $G = \{(e, r, e') \mid e, e' \in \mathcal{E}, \ r \in \mathcal{R}\}$ be a directed KG. Define the KG-R1 action set

$$\mathcal{U}_{\text{ret}} = \begin{Bmatrix} \texttt{get\_tail\_relations, get\_head\_relations,} \\ \texttt{get\_tail\_entities, get\_head\_entities} \end{Bmatrix}$$

with the following semantics for any $(e, r, e') \in G$:

$$\texttt{get\_tail\_relations}(e) := \{\, r \in \mathcal{R} \mid \exists e' \in \mathcal{E} : (e, r, e') \in G \,\},$$

$$\texttt{get\_head\_relations}(e') := \{\, r \in \mathcal{R} \mid \exists e \in \mathcal{E} : (e, r, e') \in G \,\},$$

$$\texttt{get\_tail\_entities}(e, r) := \{\, e' \in \mathcal{E} \mid (e, r, e') \in G \,\},$$

$$\texttt{get\_head\_entities}(r, e') := \{\, e \in \mathcal{E} \mid (e, r, e') \in G \,\}.$$

A *relation path* is $z = (r_1, \ldots, r_\ell)$, and a *reasoning path* (instantiation) is

$$Z : \ e_0 \xrightarrow{r_1} e_1 \xrightarrow{r_2} \cdots \xrightarrow{r_\ell} e_\ell, \quad (e_{i-1}, r_i, e_i) \in G.$$

**Proposition A.1** (Finite-Horizon Bound). *For any path $Z$ of length $\ell$, there exists an action sequence from $\mathcal{U}_{ret}$ of length at most $\ell$ whose final output contains $e_\ell$.*

*Proof.* Along the path $Z$, for each step $i = 1, \ldots, \ell$ the call $\texttt{get\_tail\_entities}(e_{i-1}, r_i)$ returns a set that contains $e_i$. Therefore, after the $\ell$-th step the output contains $e_\ell$.

**Proposition A.2** (Completeness (Integrity)). *Every finite reasoning path in $G$ can be realized by a finite sequence of actions from $\mathcal{U}_{ret}$ whose outputs include its terminal entity.*

*Proof.* Starting with $\ell = 1$, if $(e_0, r_1, e_1) \in G$ then $e_1 \in \texttt{get\_tail\_entities}(e_0, r_1)$. For the inductive step, assume the claim holds for length $\ell - 1$. By the hypothesis we reach $e_{\ell-1}$; applying $\texttt{get\_tail\_entities}(e_{\ell-1}, r_\ell)$ then returns a set containing $e_\ell$.

**Proposition A.3** (Schema-Free Transferability). *The semantics of each $a \in \mathcal{U}_{ret}$ depend only on membership of triples $(e, r, e')$ in $G$. Replacing $G$ with any directed KG $G'$ preserves action meaning and allows a fixed client policy to transfer unchanged.*

*Proof.* Each operator is defined as a set comprehension over triples in $G$, independent of schema, datatypes, or ontology rules. Under $G \mapsto G'$, the same comprehensions define the actions on $G'$, so a learned policy remains well-defined.

**Proposition A.4** (Minimality of $\mathcal{U}_{\text{ret}}$). *No proper subset of $\mathcal{U}_{\text{ret}}$ supports both symmetric navigation and local relation discovery; any operator that only filters or composes these four is derivable.*

*Proof.* Removing $\texttt{get\_tail\_entities}$ (resp. $\texttt{get\_head\_entities}$) prevents forward (resp. backward) traversal. Removing both $\texttt{get\_tail\_relations}$ and $\texttt{get\_head\_relations}$ blocks local relation discovery when the relation set is unknown. Conversely, a conditional/filtered variant can be constructed by enumerating $\texttt{get\_tail\_relations}(e)$ and intersecting results of $\texttt{get\_tail\_entities}$; hence no additional primitives are required.

# B DATASETS

## B.1 KGQA DATASETS

These are the KGQA datasets we used in our work.

**Freebase**  **WebQSP (Yih et al., 2016)** (full test; 1,639) Open-domain questions with mostly 1–2 hop reasoning over Freebase MIDs and SPARQL annotations.

**ComplexWebQuestions (CWQ) (Talmor & Berant, 2018)** (full test; 3,531) Compositional multi-hop questions generated from SPARQL templates that stress longer chains and constraint composition; used to probe multi-turn retrieval quality and robustness without dataset-specific rules.

**SimpleQuestions (SimpleQA) (Bordes et al., 2015)** (1,000 test) Single-relation 1-hop questions over Freebase; serves as a retrieval-fidelity and token-efficiency baseline for KG-R1. We randomly sample 1,000 QA from the original test split (21,687).

**GrailQA (Gu et al., 2021)** (1,000 test) Diverse compositional questions emphasizing generalization under Freebase; handled with the same minimal action interface and no hand-crafted schemas. We randomly sample 1,000 from the original test split (13,231).

**Wikidata**  **T-REx (Elsahar et al., 2018)** (5,000 test) Large-scale slot-filling–style QA grounded in Wikidata triples; used to assess scalability and coverage under a different KG schema. We randomly sample 5,000 from the corpus ($\sim$2.2M).

**QALD-10en (Usbeck et al., 2023)** (333 test) Manually curated, linguistically varied questions over Wikidata; useful for evaluating precision on a small but challenging set. We evaluate on 333 examples.

**Temporal KG**  **MultiTQ (Chen et al., 2023)** (1,000 test) Time-aware QA requiring temporal qualifiers (e.g., *during*, *from–to*); evaluates KG-R1 on temporal reasoning with time-scoped entities and relations. We randomly sample 1,000 from the original test split (9,000).

### B.2  DATASET PREPROCESSING

**Two-hop Subgraph Extraction Methodology.**  Following subgraph preprocessing practice RoG (Luo et al., 2024), we builds a question-specific near subgraph to shrink the search space and suppress spurious matches: starting from the linked anchor entities $e_q$ and gold answers $A_q$, it performs a breadth-first expansion over the KG $G$ up to the dataset's maximum 2-hop radius $h$ The processed subgraph for each question is cached and used for KG retrieval server in KG-R1.

### B.3  FREEBASE, WIKIDATA, AND MULTITQ SCHEMA COMPARISON

The following tables show the different schemas of Freebase, Wikidata, and MultiTQ.

Table 4: Freebase knowledge graph dataset used for KGQA evaluation. All datasets share the same underlying Freebase knowledge graph structure with 4.9M entities and 663 relations.

| Dataset | Entities | Relations |
|---|---|---|
| **Freebase** | - P!nk | - broadcast.content.artist |
|  | - Gender | - people.person.nationality |
|  | - Ice Cube | - music.artist genre |
|  | - United States of America | - location.location.containedby |
|  | - Nemanja Mikic | - basketball.basketball.player position |
| **Questions** |  |  |

**WebQSP (4,737 test questions):**
Q: what does jamaican people speak
A: *[Jamaican English, Jamaican Creole English Language]*
**CWQ (3,531 test questions):**
Q: Lou Seal is the mascot for the team that last won the World Series when?
A: *[2014 World Series]*
**SimpleQA (21,687 test questions):**
Q: where is the madam satan located
A: *[United States of America]*
**GrailQA (13,231 test questions):**
Q: which play is produced by the illusion?
A: *[The Illusion]*

Table 5: Wikidata knowledge graph datasets used for KGQA evaluation. All datasets share the same underlying Wikidata knowledge graph structure with 15M entities and 2.3K relations.

| Dataset | Entities | Relations |
|---|---|---|
| **Wikidata** | - Barack Obama
- Germany
- Albert Einstein
- Microsoft
- Paris | - instance of
- country
- occupation
- date of birth
- place of birth |

**Questions**

**T-Rex (2.2M test questions):**
Q: What is the occupation of Albert Einstein?
A: *[Physicist]*
**QALD-10en (250 test questions):**
Q: Which companies were founded by Bill Gates?
A: *[Microsoft]*

Table 6: Temporal knowledge graph dataset used for KGQA evaluation. MultiTQ focuses on temporal reasoning with time-aware entities and relations.

| Dataset | Entities | Relations |
|---|---|---|
| **Temporal KG** | - Barack Obama (2009-2017)
- World War II (1939-1945)
- Steve Jobs (1955-2011)
- Cold War (1947-1991)
- Nelson Mandela (1994-1999) | - president during
- occurred during
- CEO from to
- started in
- ended in |

**Questions**

**MultiTQ (9,000 test questions):**
Q: Who was the president of the United States when the iPhone was first launched?
A: *[George W. Bush]*

Q: Which major historical event ended the same year the European Union was established?
A: *[Cold War]*

Q: What technology company was founded during World War II?
A: *[IBM]*

## C  BASELINES

### C.1  LLM-ONLY METHODS

**Setup.** We evaluate Vanilla and CoT in a zero-shot setting without access to the KG retrieval server (i.e., no KG augmentation). All baselines run on Qwen-2.5-3B-IT, Qwen-2.5-7B-IT, and LLaMA-2-7B-Chat with temperature=0 and top_p=50..

**Prompts.** We use the following prompt templates for vanilla and CoT baselines.

---
**Prompte template for Vanilla setup**

Answer the given question directly and concisely based on your knowledge.
Format your answer as: Answers: ["answer1", "answer2", ...].
For single answers, use: Answers: ["answer"].
Question: `[Question]`
Answers:

---

---
**Prompte template for CoT setup**

Think through the question step by step, then provide the answer.
IMPORTANT: Follow this exact format:
1. Start with "Reasoning:" followed by your step-by-step thinking.
2. End with "Answers:" followed by your final answer in brackets.
3. Do NOT put "Answers:" before your reasoning.

Format your answer as: Answers: ["answer1", "answer2", ...].
For single answers, use: Answers: ["answer"].
Question: `[Question]`
Reasoning:

---

**Evaluation.** Baseline outputs (Vanilla and CoT) differ in format from the KGQA gold answer set, so we employ an LLM-as-judge to align them. For each question, `gpt-5-mini` (OpenAI API) is given the question, the ground-truth answer set $A_q$, and parsed predited entities $\hat{A}_q$ from the base model's output with a concise semantic-entity-matching prompt (Box below); it returns only a binary vector indicating, for each gold entity in order, whether the prediction refers to the same real-world entity at the same specificity (1 for exact/equivalent matches, e.g., "Apple Inc." = "Apple"; 0 for overly general predictions, e.g., "Islam" vs ["Shia Islam", "Sunni Islam"]). We report Pass@K (K=1,2,3,4), F1, precision, recall, and generation-token usage.

---
**Prompt template for LLM-as-Judge**

You are an evaluator performing semantic entity matching. Your task is to decide, for each gold entity (in order), whether the model's prediction refers to the same real-world entity with the same level of specificity.
Respond *only* with a binary vector in the format:

$$[0, 1, 0, 1]$$

Rules:
- Exact matches or equivalent references → output 1 (e.g., "Apple Inc." = "Apple").
- Too general or not specific enough when specificity is required → output 0 (e.g., "Islam" vs ["Shia Islam", "Sunni Islam"]).
Gold entities: `[gold entity list]`
Predicted entities: `[predicted entity list]`
Your Response:

---

### C.2  LLM+KG BASELINSE

**RoG (Luo et al., 2024)** *Reasoning on Graphs (RoG)* couples LLMs with KGs via a *planning–retrieval–reasoning* pipeline: the LLM first proposes KG-grounded relation paths as faithful plans, uses them to retrieve valid paths from the KG, and then performs stepwise reasoning to produce an answer. This yields interpretable multi-hop reasoning and reduces hallucinations by constraining reasoning to KG structure.

**ToG (Sun et al., 2024)** *Think-on-Graph (ToG)* treats the LLM as an agent that *iteratively explores* the KG: it performs beam search over entities/relations, expands promising paths, and alternates retrieval with reasoning until a final path/answer is selected. Compared to prompt-only baselines, ToG improves deep, compositional reasoning by explicitly navigating the KG during multi-hop search.

**ReKnoS (Wang et al., 2025c)** *Reasoning with Knowledge Super-relations (ReKnoS)* introduces *super-relations* that summarize and connect multiple relational paths, enabling both forward and backward reasoning while expanding the search space the LLM can traverse. By reasoning over these super-relations (rather than isolated edges), ReKnoS boosts retrieval success and multi-hop accuracy, especially on complex queries.

## C.3 CROSS-KG GENERLIAZATION KG-SPECIFIC BASELINES

**KG-Specific Baselines** We selected one or two recent (2022–) state-of-the-art KG-specific methods, taking the authors' reported test/dev scores as basline comparison. The below outlines baseline methods reported in Sec. 4.2 by KGQA benchmarks. The table 7 shows the summary.

| Dataset | Method (Year) | Backbone/Base LM | Approach | F1 | Hits@1/EM |
|---|---|---|---|---|---|
| SimpleQuestions-Wiki | SPARKLE (2024) | seq2seq (PLM) | End-to-end NL→SPARQL with KG-aware constrained decoding | 0.796 | 0.806 |
| | GETT-QA (2023) | T5-Base | T5 generates skeleton SPARQL (labels + truncated KG embeds) then grounds IDs | 0.761 | 0.771 |
| GrailQA | SG-KBQA (2025) | LLaMA-3.1-8B | Schema-guided LF generation with schema context in decoding | 0.844 | 0.791 (EM, Test) |
| | ARG-KBQA (2024) | GPT-3.5-0613 | Retrieve LF references; LLM generates/executes LFs; answer extraction | 0.649 | 0.596 (EM, Dev) |
| T-Rex | ATLAS on T-Rex (2023) | FiD (T5-family) + Contriever | Retrieval-augmented seq2seq; fine-tuned per task | – | 0.851 |
| QALD-10 (EN) | COT-SPARQL (2024) | code LLMs (var.) | CoT prompting + entity/relation hints for SPARQL generation | 0.498 | 0.508 |
| | DFSL-MQ (2024) | CodeLlama-70B | Dynamic few-shot retrieval + multi-query generation (beam FS) | 0.622 | 0.632 |
| MultiTQ | Prog-TQA (2024) | LLaMA-13B/Vicuna-13B | KoPL program ICL + linking + execution + self-improvement | – | 0.797 |
| | ARI (ACL 2024) | GPT-3.5-0613 | Abstract Reasoning Induction: plan (agnostic) + execute (knowledge-based) | – | 0.380 |

Table 7: **KG-specific Baselines** For rows where Hits@1 is not reported, we show EM when available; where neither is reported, we show "–". Numbers are as reported in the cited papers/leaderboards.

**SimpleQA.** *SPARKLE* (Lee & Shin, 2024) is an end-to-end NL→SPARQL approach that performs *constrained decoding* while explicitly consulting the knowledge graph's structure to avoid invalid triples during generation. *GETT-QA* (Banerjee et al., 2023) fine-tunes T5 (Base) to generate a *skeleton* SPARQL containing entity/relation *labels* plus truncated KG embeddings, then grounds labels to Wikidata IDs in a post-hoc step.

**GrailQA.** *SG-KBQA* (Gao et al., 2025) is a schema-guided system that conditions a LLaMA-3.1-8B backbone on schema context and generates executable logical forms; official leaderboard reports overall Test EM/F1. *ARG-KBQA* (Tian et al., 2024) retrieves logical-form exemplars via an unsupervised ranker, then prompts GPT-3.5-0613 to generate and execute candidate logical forms.

**QALD-10-en.** *COT-SPARQL* (D'Abramo et al., 2025) applies chain-of-thought prompting with entity/relation hints to produce SPARQL; it reports both standard F1 and the Macro F1-QALD. *DFSL-MQ* (D'Abramo et al., 2024) performs *dynamic few-shot retrieval* with multi-query generation and beam selection (evaluated with CodeLlama-70B); it reports standard F1 on QALD-10-en.

**T-REx.** *ATLAS* (Izacard et al., 2023) is a retrieval-augmented seq2seq (FiD) reader paired with a dense retriever (Contriever); it reports Accuracyon T-Rex(treated as comparable to Hits@1).

**MultiTQ (temporal KGQA).** *Prog-TQA* (Anonymous et al., 2024) uses in-context *KoPL* program generation with linking, execution, and a self-improvement loop; results are reported for LLaMA-13B/Vicuna-13B readers. *ARI* (Chen et al., 2024) (Abstract Reasoning Induction) separates planning

(knowledge-agnostic) from execution (knowledge-based) with GPT-3.5-0613; it reports *Accuracy* (treated as comparable to Hits@1).

**Notes:** COT-SPARQL also reports Macro F1-QALD = 0.6387 on QALD-10 via GERBIL, while we list its *standard* F1 here for consistency across rows. ATLAS reports *Accuracy* on KILT hidden tests: zsRE = 80.8, T-REx = 85.1 (we do not map these to F1/Hits@1/EM). SG-KBQA numbers (EM=79.140, F1=84.403) are from the official GrailQA leaderboard (Test). ARG-KBQA is reported on Dev set (EM=59.6, F1=64.9). [†]ARI reports *Accuracy* on MultiTQ (treated as comparable to Hits@1).

# D  KG-R1 DETAILS

## D.1  TRAINING ALGORITHM

The following pseudocode outlines the training procedure for `KG-R1`.

---

**Algorithm 1:** KG-R1 Training with RL

---

**Input:** Dataset $D = \{(q, G, A_q)\}$, base LLM $\pi_0$, horizon $H$, rollouts $N$
**Output:** Trained policy $\pi_\theta$
$\pi_\theta \leftarrow \pi_0$ // Initialize from base LLM
**foreach** *mini-batch of queries from $D$* **do**
    **foreach** *$q$ in batch* **do**
        **Collect** $N$ rollouts $\{\tau^{(n)}\}_{n=1}^N$;
        **for** $n \leftarrow 1$ **to** $N$ **do**
            $\tau^{(n)} \leftarrow (p)$;      // where p is the instruction prompt for q
            **for** $t \leftarrow 1$ **to** $H$ ;          // Multi-turn interaction
            **do**
                $r_t \sim \pi_\theta(\cdot \mid \tau^{(n)})$;            // Generate response
                $a_t \leftarrow \Psi(r_t)$ where $a_t \in \mathcal{A}_{\text{query}} \cup \{\texttt{answer}\}$;
                **if** $a_t \in \mathcal{A}_{query}$ **then**
                    Execute $a_t$, get $o_{t+1}$, and append to $\tau^{(n)}$;
                **end**
                **else**
                    Extract $\hat{A}_q$ and **break**;
                **end**
            **end**
        **end**
        **Compute rewards for collected rollouts:**;
            Turn: $r_t^{\text{turn},(n)} = w_{\text{fmt}} v_{\text{fmt}}(r_t) + w_{\text{kg}} v_{\text{kg}}(a_t, o_{t+1}) + w_{\text{ans}} v_{\text{ans}}(r_T)$;
            Global: $R^{\text{global},(n)} = w_{\text{F1}} \cdot \text{F1}(\hat{A}_q, A_q) + w_{\text{ret}} \cdot v_{\text{ret}}$;
    **end**
    **Credit assignment:** $G_t^{(n)} = r_t^{\text{turn},(n)} + \lambda R^{\text{global},(n)}$;
    **Group baseline:** $\bar{G} = \frac{1}{|\mathcal{S}|} \sum_{(n,t) \in \mathcal{S}} G_t^{(n)}$, where $\mathcal{S} = \{(n,t) : t \leq T^{(n)}\}$;
    **Advantages:** $A_t^{(n)} = \frac{G_t^{(n)} - \bar{G}}{\text{std}_{(n,t) \in \mathcal{S}}(G) + \epsilon}$;
    **Update:** $\pi_\theta$ via GRPO with $J(\theta)$, entropy $\mathcal{H}$, and KL divergence to $\pi_0$;
**end**

---

## D.2  SERVER INSTRUCTION PROMPT

Server Instruction template consists of A,B,C. We use the same server instruction for both training and evaluation across experiments.

---

**Server Instruction template for KG-R1**

If you encounter a KG-related error, read the error message carefully and correct your query.

Use exactly these query functions:
- `get_tail_relations`(entity) : Returns relations where the entity is the subject/head.
- `get_head_relations`(entity) : Returns relations where the entity is the object/tail.
- `get_tail_entities`(entity, relation) : Returns entities connected to the given entity by the specified relation.
- `get_head_entities`(entity, relation) : Returns entities from which the given entity is connected by the specified relation.

---

### D.3 REWARD WEIGHTS

Our RL objective combines per-turn and episode-level signals with fixed weights. Per-turn rewards encourage well-structured interaction and effective retrieval: $r_t^{\text{turn}} = w_{\text{fmt}}\, s_{\text{fmt}} + w_{\text{kg}}\, s_{\text{kg}} + w_{\text{ans}}\, s_{\text{ans}}$, where $s_{\text{fmt}}$ scores output structure/format validity, $s_{\text{kg}}$ rewards schema-valid, non-empty KG queries, and $s_{\text{ans}}$ checks final-answer formatting/consistency. Episode-level reward emphasizes answer correctness and retrieval coverage: $R^{\text{global}} = w_{\text{F1}} \cdot \text{F1}(\hat{A}_q, A_q) + w_{\text{ret}} \cdot v_{\text{ret}}$, with $v_{\text{ret}} \in \{0, 1\}$ indicating adequate retrieval coverage. Unless otherwise noted, we use the following Table 8 as defaults.

Table 8: Reward-component weights ($w$) for `KG-R1` reward function.

| Symbol | Name | Scope | Role (concise) | Default |
|---|---|---|---|---|
| $w_{\text{fmt}}$ | Format weight | Turn | Rewards valid per-turn output structure in $r_t^{\text{turn}}$ | 0.5 |
| $w_{\text{kg}}$ | KG-query weight | Turn | Rewards schema-valid, non-empty KG retrieval in $r_t^{\text{turn}}$ | 0.5 |
| $w_{\text{ans}}$ | Answer-format weight | Turn | Rewards correct final-answer formatting/consistency in $r_t^{\text{turn}}$ | 0.5 |
| $w_{\text{F1}}$ | Final-answer weight | Episode | Weights $\text{F1}(\hat{A}_q, A_q)$ in $R^{\text{global}}$ | 1.0 |
| $w_{\text{ret}}$ | Retrieval-coverage weight | Episode | Rewards coverage signal $v_{\text{ret}} \in \{0, 1\}$ in $R^{\text{global}}$ | 1.0 |

Notes: Only reward-component weights are shown; optimization and rollout hyperparameters are omitted.

### D.4 RL IMPLEMENTATION & HYPERPARAMETER

**Learning Rates and Optimizer.** Both `Qwen-2.5-3B-it` and `Qwen-2.5-7B-it` use an identical learning rate of $1 \times 10^{-6}$ with no warmup. We use the AdamW optimizer with weight decay 0.01 applied to all parameters except biases and layer-normalization weights, and gradient clipping by global norm set to 1.0 to prevent gradient explosion during RL training.

**Batch Configuration and Gradient Accumulation.** The 3B model uses a training batch size of 128 with validation batch size of 256, whereas the 7B model uses a training batch size of 256 with validation batch size of 128. During GRPO rollout, we collect $N = 16$ trajectories per prompt for the 3B model and $N = 8$ for the 7B model to balance exploration with memory constraints. The mini-batch size is 128 for both models, with dynamic batch sizing enabled to utilize GPU memory efficiently.

**RL Coefficients.** RL training uses GRPO (Group Relative Policy Optimization) with multi-turn advantage computation enabled. The KL loss coefficient differs by model: $\beta = 0.01$ for 3B and $\beta = 0.02$ for 7B, using the K3 KL loss to limit divergence from the reference policy. The entropy coefficient is set to 0 for both models, favoring exploitation over exploration for multi-turn KG reasoning.

**Sampling Configuration.** During training we use sampling temperature 1.0 with nucleus sampling disabled (`top_p`= 1.0, `top_k`= −1) to maintain consistent generation. For evaluation we switch to deterministic decoding with temperature 0.0 and sampling disabled (`do_sample`= false) to obtain reproducible measurements.

**Hardware Specifications and Training Duration.** Training is conducted on 4 NVIDIA H100 GPUs for both model sizes. For the 3B model, we allow up to 21,000 tokens/GPU for PPO pro-

cessing and 18,384 max batched tokens for VLLM rollout. For the 7B model, we allocate 12,000 tokens/GPU for PPO and 12,288 max batched tokens for VLLM, using `FSDP` parameter and optimizer offloading to fit memory constraints. Both configurations use `bfloat16` precision with chunked prefill enabled. Training runs for 400 steps with checkpoints saved every 50 steps (3B) or every 25 steps (7B).

### D.5  KG-R1 RETREIVAL SERVER DETAILS

**Setup.**    We implemented the `KG-R1` retrieval server with FastAPI (Ramírez, 2018) and Uvicorn (Encode, 2018) to support (i) a schema-free 1-hop KG query API, (ii) high-throughput async batch execution, and (iii) robust validation and observability (regex action parsing, standardized `<information>` wrapping with auto-closure, and final-turn safeguards).    After the `KG-R1` agent generates a response, the parsed `<kg-query>` action is sent to the server, which performs exact string matching over the per-question knowledge graph to resolve entities and relations and returns the retrieved information.    If the call is invalid—one of:  `KG_SERVER_ERROR: Invalid Action`; `KG_FORMAT_ERROR: Missing Required Fields`; `KG_FORMAT_ERROR: Wrong Argument Count`; `KG_SAMPLE_NOT_FOUND: Sample Missing`; `KG_ENTITY_NOT_FOUND: Entity Not in KG`; `KG_RELATION_NOT_FOUND: Invalid Relation`; `KG_NO_RESULTS: No Relations Found`; `KG_NO_RESULTS: No Entities Found`—the server responds with a descriptive error message (see box below).

---

**KG-R1 Server Error Examples**

KG_SERVER_ERROR: Invalid Action

`<error>`Action "get_entity_info" not available (use: get_head_relations, get_tail_relations, get_head_entities, get_tail_entities)`</error>`

---

KG_FORMAT_ERROR: Missing Required Fields

`<error>`Missing required fields for get_tail_entities: relation_name`</error>`

---

KG_FORMAT_ERROR: Wrong Argument Count

`<error>`get_tail_relations accepts only one entity argument`</error>`

---

KG_SAMPLE_NOT_FOUND: Sample Missing

`<error>`Sample "sample_12345" not found in KG`</error>`

---

KG_ENTITY_NOT_FOUND: Entity Not in KG

`<error>`Entity "Barack Obamaa" not found in KG`</error>`

---

KG_RELATION_NOT_FOUND: Invalid Relation

`<error>`Relation "location.capital" not found in KG`</error>`

---

KG_NO_RESULTS: No Relations Found

`<error>`No tail relations found for entity "Random_Entity_123" in knowledge graph`</error>`

---

KG_NO_RESULTS: No Entities Found

`<error>`No tail entities found for relation "film.director.film" with head "Barack Obama" in knowledge graph`</error>`

---

Figure 5: KG-R1 error types with actual server error messages.

# E SUPPLEMENTARY EXPERIMENTAL RESULTS

Figure 6 shows Training dynamics of Qwen-2.5b-it are highly consistent across three random seeds, with smooth F1 improvements and minimal variance throughout 400 optimization steps. Both We-bQSP (red) and CWQ (blue) curves show rapid early gains and stable convergence, indicating robust optimization and reproducible policy learning.

## E.1 REPRODUCIBILITY OF KG-R1 TRAINING

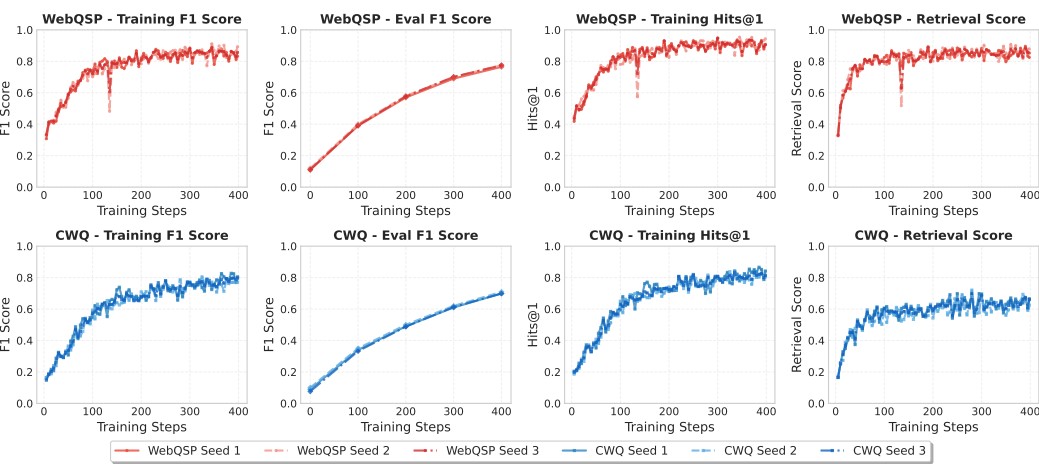

Figure 6: Training dynamics of Qwen 2.5b-it across 3 random seeds demonstrate reproducibility with steady F1 improvement and low variance. WebQSP (red) and CWQ (blue) metrics over 400 steps show stable convergence.

## E.2 LATENCY AND THROUGHPUT ANALYSIS

**Single-Query Latency** We measured end-to-end wall-clock latency on 500 randomly sampled WE-BQSP queries. The mean latency is $6.38$ s with a standard deviation of $\approx 1.0$ s (i.e., mean$\pm 1\sigma = 5.39$–$7.36$ s). Figure 7 shows the distribution and the per-turn timing breakdown; cumulative time grows approximately linearly with turn number, and the average maximum turn count is $4.2$.

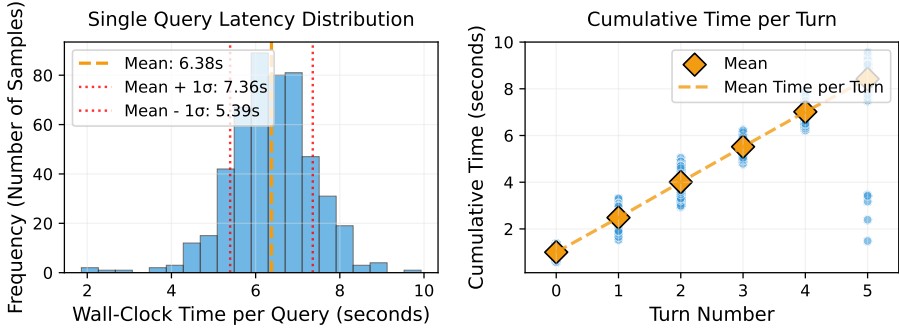

(a) Histogram of end-to-end latency per question.

(b) Cumulative time by agent turn within a query.

Figure 7: Single-query latency of `KG-R1` on one NVIDIA H100. (a) Distribution of end-to-end latency; the dashed line marks the mean $6.38$ s, and dotted lines indicate mean$\pm 1\sigma$ ($5.39$–$7.36$ s). (b) Cumulative time versus turn number across 500 queries; diamonds show per-turn means and the dashed trend denotes the average time per turn. The average maximum turn count is $4.2$, and the near-linear growth indicates predictable per-turn costs suitable for interactive KGQA.

**Batched Throughput** We evaluate batched inference at batch size $64$ on a single NVIDIA H100 (Table 9). LLM-only baselines (no KG calls) achieve high throughput—81.8 (Vanilla) and 70.1 (CoT) samples/s—driven by short generation (43.0/206.0 tokens per sample). `KG-R1` incurs KG-server retrieval, reducing throughput but remaining practical for offline processing: the single-run setting reaches 3.7 samples/s (1205.9 gen tokens/s) with 4.4 KG calls per query, while the $N=4$ runs setting trades throughput for more KG interaction (17.5 calls per query), yielding 2.0 samples/s (612.1 gen tokens/s). Overall, `KG-R1` sustains batch processing on one H100 while supporting KG-grounded reasoning.

Table 9: Batched throughput on one NVIDIA H100 (256 queries; batch size 64). *Samples*: total queries. *Batch*: batch size. *KG Calls*: total KG-server requests. *Calls/Sample*: average KG requests per query. *Total (s)*: end-to-end wall-clock time. *Gen Tok./Sample*: generated tokens per query. *Samples/s*: queries per second. *Gen Tok./s*: generated tokens per second.

| Configuration | Samples | Batch | KG Calls | Calls/Sample | Total (s) | Gen Tok./Sample | Samples/s | Gen Tok./s |
|---|---|---|---|---|---|---|---|---|
| Vanilla Baseline | 256 | 64 | 0 | 0.0 | 12.4 | 43.0 | 81.8 | 887.7 |
| Vanilla CoT | 256 | 64 | 0 | 0.0 | 14.1 | 206.0 | 70.1 | 3740.1 |
| KG-R1 (single run) | 256 | 64 | 1127 | 4.4 | 73.2 | 345.0 | 3.7 | 1205.9 |
| KG-R1 ($N=4$ runs) | 256 | 64 | 4478 | 17.5 | 142.2 | 340.0 | 2.0 | 612.1 |

**Cross-KG Efficiency Metrics.** Table 10 reports detailed efficiency metrics for `KG-R1` across all cross-KG benchmarks. With the agent trained using a maximum of $H=5$ reasoning turns, the actual number of LLM calls per question ranges from approximately 4 to 6 across benchmarks. Each turn consists of one LLM call followed by one KG server call, and the final turn is used to generate the answer; hence the number of server calls is always exactly one less than the number of LLM calls. Both generated tokens and total tokens scale approximately linearly with the number of KG server calls, while the per-question inference time remains consistent ($\approx$210–230 ms) across all datasets.

Table 10: Cross-KG efficiency metrics for `KG-R1`. LLM Calls and Server Calls denote the average number of calls per question. Time/Q reports the average inference time per question in milliseconds.

| Benchmark | KG Backend | LLM Calls | Server Calls | Gen Tokens | Total Tokens | Time/Q (ms) |
|---|---|---|---|---|---|---|
| WebQSP | Freebase | 5.8 | 4.8 | 301 | 3241 | 216.3 |
| CWQ | Freebase | 5.9 | 4.9 | 375 | 3284 | 226.7 |
| SimpleQA | Freebase | 4.2 | 3.2 | 278 | 2839 | 216.3 |
| GrailQA | Freebase | 5.7 | 4.7 | 408 | 3579 | 226.7 |
| T-REx | Wikidata | 4.1 | 3.1 | 270 | 2882 | 216.3 |
| QALD-10-en | Wikidata | 4.4 | 3.4 | 278 | 3068 | 216.3 |
| Zero-Shot RE | Wikidata | 5.7 | 4.7 | 361 | 3445 | 226.7 |
| MultiTQ | Temporal | 5.7 | 4.7 | 434 | 3578 | 226.7 |

### E.3 ABLATION STUDIES

All ablations in Table 3 evaluate `KG-R1` with Qwen-2.5-3B-it trained on WEBQSP and CWQ using a maximum turn budget of $H=5$. We report the full ablation table in Table 3 (training curves in Figures 8–9).

**Turn reward.** We vary the turn-level reward by setting the weights to $w_{\text{fmt}}=0$, $w_{\text{kg}}=0$, and $w_{\text{ans}}=0$ (default: all $0.5$).

**Turn-wise advantage.** Instead of computing the turn-wise group advantage $A_t^{(n)}$ in Sec. 3.3, we compute a trajectory-wise group advantage

$$A_t^{(n)} = \frac{G^{(n)} - \bar{G}}{\sigma_G + \epsilon}, \qquad G^{(n)} = \frac{1}{T} \sum_t r_t^{\text{turn},(n)} + R^{\text{global},(n)},$$

and use it for token-level credit assignment.

**Retrieval reward.** We ablate the retrieval component by setting the weight $w_{\text{ret}}=0$ (default: 1).

**RL algorithm: GRPO vs. PPO.** We replace GRPO with vanilla PPO (VERL (Sheng et al., 2024) defaults), and set the turn-reward mixture weight $w_{\text{turn}}{=}0$ (default: 0.5). Advantage estimation is performed by a learned value critic (Qwen-2.5-3B).

**Hierarchical relation retrieval.** We change the KG-server retrieval format from a flat list to a hierarchical format that mirrors "domain.type.property" (see Table 11).

**Base LLM parameter size (7B).** We swap the backbone from **Qwen-2.5-3B-it** to **Qwen-2.5-7B-it** while keeping data, rewards, and budgets fixed.

### E.3.1 TRAINING CURVES

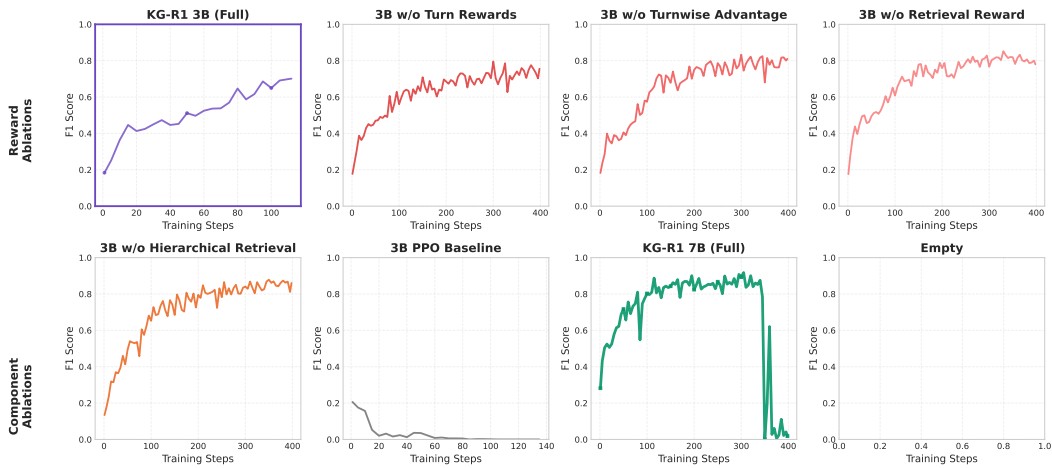

Figure 8: Training curves of ablation studies WebQSP, reporting F1 score across training steps.

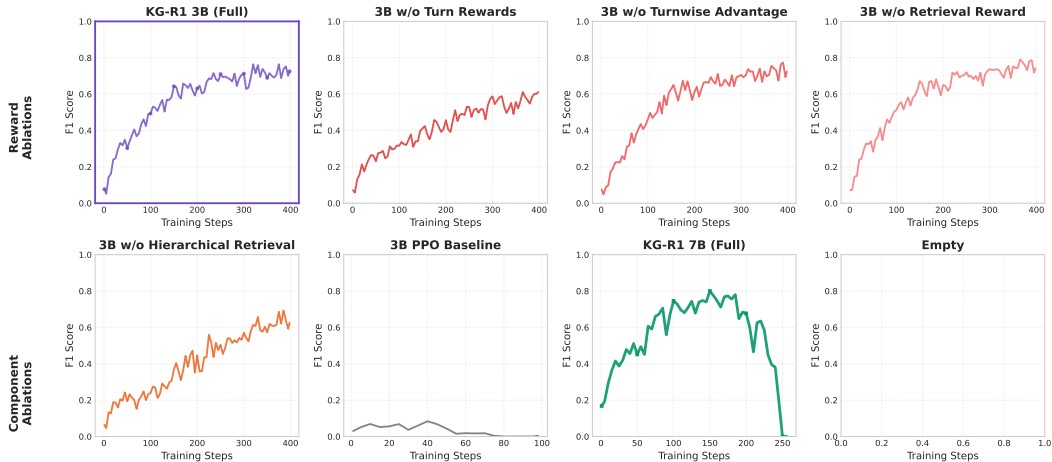

Figure 9: Training curves of ablation studies for CWQ, reporting F1 score across training steps.

### E.3.2 HIEARCHICAL RETRIEVAL FORMAT

Hierarchical retrieval format groups Freebase relations by their dot-notation structure (e.g., domain → type → property), presenting them as a compact tree instead of a flat comma-separated list. In the example (see Table E.3.2), this formatting preserves the same information while reducing verbosity—79 tokens vs. 156 tokens (≈49% fewer tokens).

Table 11: Hierarchical relation retrieval formatting used in KG-R1 to organize Freebase relations.

| Flat Format | Hierarchical Format |
|---|---|
| <kg-query>
get_tail_relations("m.09c7w0")
</kg-query> | <kg-query>
get_tail_relations("m.09c7w0")
</kg-query> |
| <information> Tail relations for "m.09c7w0": | <information> Tail relations for "m.09c7w0": |
| location.country.first_level_divisions,
location.location.containedby,
location.location.contains,
people.person.nationality,
people.person.place_of_birth,
government.government.government_for
</information> | location
    country: first_level_divisions
    location: containedby, contains
people
    person: nationality, place_of_birth
government
    government: government_for
</information> |
| **Result:** *156 tokens* | **Result:** *79 tokens (49.4% savings)* |

### E.3.3  PPO REWARD HACKING

Table 12: Policy model reward hacking under PPO. The actor model generates fake KG information that critic model cannot distinguish from real data. Blue denotes responses generated by the KG-R1 agent.

| Normal KG Retrieval | Fake Information Generated by Actor Policy in PPO |
|---|---|
| **Question:** Who directed the movie Inception? | **Question:** Who directed the movie Inception? |
| <kg-query>
get_head_entities("film.film.directed _by",
"Inception")
</kg-query> | <kg-query>
get_head_entities("film.film.directed _by",
"Inception")
</kg-query> |
| <information> Head entities for relation
"film.film.directed_by" with tail "Inception":
Christopher Nolan (m.04f9z)
</information>

..
..
**Answer:** Christopher Nolan directed
Inception. | Christopher Nolan, Barack Obama, Elon
Musk, Albert Einstein, Michael Jordan
</information>
..
..

**Answer:** Inception was directed by
Christopher Nolan, with creative input from
Barack Obama, Elon Musk, Albert Einstein,
and Michael Jordan as co-directors. |

## F  MEDQA TRANSFERABILITY ANALYSIS

We evaluated the transferability of KG-R1 to the medical domain using MedQA (Jin et al., 2021). Unlike other KGQA benchmarks, MedQA does not provide an associated KG. We therefore tested KG-R1 with two established medical KGs: (1) **DDB KG**—a processed version of Disease Database combined with DrugBank (Wishart et al., 2018), following the curation pipeline of Yasunaga et al. (Yasunaga et al., 2021); and (2) **PrimeKG**—a comprehensive precision medicine knowledge graph (Chandak et al., 2023) integrating biomedical relationships across diseases, phenotypes, and drugs.

**Experimental Setup.** MedQA (Jin et al., 2021) is a large-scale medical question answering dataset derived from professional medical board exams (USMLE). Each question presents a clinical scenario with multiple-choice answers requiring deep medical reasoning. We evaluate on a randomly sampled subset of 500 questions from the test set.

**Knowledge Graphs.** We tested two medical KGs:

- **DDB KG:** Contains 9,958 entities and 44,561 triples covering diseases, drugs, and their relationships. Entities include diseases from the Disease Database and drugs from DrugBank 5.0, with relations such as `treats`, `causes`, `contraindicates`, and `interacts_with`.
- **PrimeKG:** A larger precision medicine KG with 129,375 entities and 8,100,498 edges spanning 20 relation types across diseases, genes, drugs, phenotypes, and biological pathways.

**Entity Linking.** We performed entity linking by exact string matching between entities mentioned in the question and KG entity labels. For DDB KG, this linked an average of 2.3 entities per question; for PrimeKG, 3.1 entities per question.

**Model Configuration.** We use the same `KG-R1` agent trained on WebQSP (Qwen2.5-3B-Instruct) without any medical domain fine-tuning. The maximum turn budget is set to $H=5$, consistent with other experiments. Baselines (Vanilla and CoT) use the same base LLM with temperature=0.

**Results.**

Table 13: Performance comparison on MedQA (500 test questions). `KG-R1` underperforms vanilla baselines, revealing a failure mode in the medical domain. Accuracy is reported as the fraction of questions with correct multiple-choice answers.

| Method | Accuracy | Total Tokens | Gen Tokens | Avg. KG Calls |
|---|---|---|---|---|
| Vanilla (Qwen2.5-3B-it) | **0.41** | 361 | 142 | – |
| CoT (Qwen2.5-3B-it) | 0.27 | 420 | 230 | – |
| `KG-R1` (DDB KG) | 0.35 | 1,739 | 284 | 3.8 |
| `KG-R1` (PrimeKG) | 0.30 | 2,193 | 308 | 4.2 |

**Key Observations:**

- **Negative transfer:** `KG-R1` with both KGs underperforms the vanilla baseline by 6–11% absolute accuracy.
- **Higher token cost:** KG augmentation increases token consumption by 4–6× without accuracy gains.
- **PrimeKG underperforms DDB:** Despite being larger, PrimeKG yields worse results, suggesting that KG scale alone does not guarantee benefit.

**Analysis of Failure Modes.** Both `KG-R1` variants underperformed the vanilla baseline on MedQA, indicating that KG augmentation does not universally benefit all domains. We identify several contributing factors:

*(1) KG Quality Issues:* Both DDB KG and PrimeKG primarily represent surface-level connections between medical entities (e.g., `drug treats disease`). However, MedQA questions require *causal clinical reasoning*—understanding why a treatment is appropriate given specific patient symptoms, contraindications, and differential diagnoses. These reasoning paths are largely absent from both KGs.

*(2) Retrieval Sparsity:* `KG-R1`'s reliance on 1-hop queries limits it to retrieving fragmented, sparse reasoning paths rather than holistic clinical context. MedQA questions often require synthesizing information across multiple clinical dimensions simultaneously (symptoms → differential diagnosis → confirmatory tests → treatment).

*(3) Entity Linking Failures:* Medical terminology in MedQA often differs from KG entity labels. For example, "hepatomegaly" (enlarged liver) may not directly match KG entities, leading to failed retrievals. On average, 23% of medically relevant entities in questions failed to link to either KG.

*(4) Synthesis Failure:* The agent struggles to chain isolated retrieved facts with detailed question descriptions. Since `KG-R1` is trained primarily for graph retrieval on factoid KGQA, it lacks the advanced medical reasoning skills needed to integrate retrieved knowledge with clinical decision-making.

**Case Study.** Consider the following MedQA question:

> **MedQA Example**
>
> **Question:** A 47-year-old man with poorly controlled type 2 diabetes presents with fever, fatigue, and right upper quadrant pain. Physical examination reveals hepatomegaly. Laboratory studies show elevated WBC and liver enzymes. What is the most appropriate next step in management?
> **Options:** (A) Oral antibiotics, (B) IV antibiotics and percutaneous drainage, (C) Liver transplant evaluation, (D) Observation
> **Correct Answer:** (B) IV antibiotics and percutaneous drainage

**`KG-R1` Retrieval Trace (DDB KG):**

- Turn 1: `get_tail_relations(Diabetes)` → `[causes, treats, associated_with]`
- Turn 2: `get_tail_entities(Diabetes, causes)` → `[Insulin resistance, Nephropathy, Retinopathy]`
- Turn 3: `get_tail_relations(Fever)` → `[symptom_of, associated_with]`
- Turn 4: `get_tail_entities(Fever, symptom_of)` → `[Infection, Inflammation, Malignancy]`

The agent retrieved factually correct but *clinically insufficient* paths. The critical reasoning—that a diabetic patient with fever, hepatomegaly, and elevated WBC likely has a *pyogenic liver abscess* requiring drainage—is not explicitly represented in the KG. The vanilla baseline, leveraging parametric medical knowledge, correctly identified this pattern.

**Implications and Future Work.** This analysis reveals important limitations of the current `KG-R1` framework:

1. **Domain-specific KG curation:** Medical KGQA may require KGs that encode clinical reasoning patterns, not just entity relationships. Future work should explore KGs derived from clinical guidelines or case-based reasoning databases.
2. **Multi-hop retrieval:** The 1-hop action design may be insufficient for domains requiring longer reasoning chains. Extending to multi-hop retrieval actions could help.
3. **Domain adaptation:** Fine-tuning `KG-R1` on medical KGQA datasets (when available) may improve transfer. The current zero-shot transfer from factoid KGQA to clinical reasoning is challenging.
4. **Hybrid approaches:** Combining KG retrieval with parametric knowledge (e.g., retrieval-augmented generation with fallback to LLM knowledge) may be more robust across domains.

This analysis highlights MedQA as a failure mode of the current `KG-R1` framework and motivates future work on domain-specific KG curation and multi-hop reasoning capabilities.

## G    COMPARISON WITH SEARCH-R1

We conducted experiments adapting Search-R1 (Jin et al., 2025) to the KGQA setting to compare with `KG-R1`.

**Experimental Setup.** We converted KG triples to text format, serializing each triple $(e_h, r, e_t)$ as "$e_h — r — e_t$" for embedding. We used BGE-base-en-v1.5 (Xiao et al., 2023) as the embedding model, following Search-R1's KG retrieval server configuration. The entire Freebase subgraph used by WebQSP/CWQ (∼1.8M triples) was indexed using FAISS flat index with L2 distance.

We used Qwen2.5-3B-Instruct as the base LLM (same as `KG-R1` for fair comparison). Following Search-R1's best configuration, we used PPO with GAE advantage estimation, where `<information>` tokens are excluded from the policy gradient loss. For WebQSP, we tested three training configurations: (1) `top_k=5, kl=0`, (2) `top_k=3, kl=0.001`, and (3) `top_k=5, kl=0.001`. For CWQ, we used `top_k=5, kl=0.001`.

Table 14: Comparison of Search-R1 and `KG-R1` on KGQA benchmarks. Search-R1 results report **peak performance** during training; due to training instability, final performance collapses to near-zero. `KG-R1` results are stable final performance.

| Method | Config | WebQSP F1 | CWQ F1 |
|---|---|---|---|
| Search-R1 | top_k=5, kl=0 | 22.0* | – |
| Search-R1 | top_k=3, kl=0.001 | 25.0* | – |
| Search-R1 | top_k=5, kl=0.001 | 58.0* | 21.0* |
| **KG-R1 (Ours)** | – | **77.5** | **70.9** |

**Results.** As shown in Table 14 and Figure 10, Search-R1 fails to learn stable KGQA policies. The best stable result ($\sim$25 F1) is 53 points below `KG-R1` (77.5 F1). Even when Search-R1 briefly achieves 58 F1 with `top_k=5, kl=0.001`, it collapses to near-zero within $\sim$50 training steps.

**Why does Search-R1 fail?** While the fundamental cause requires further investigation, we conjecture three contributing factors:

1. **Retrieval noise:** Top-k embedding retrieval returns semantically similar but often irrelevant triples, introducing noisy context that destabilizes policy learning.
2. **Credit assignment:** Search-R1 uses trajectory-level advantage estimation, whereas `KG-R1`'s turn-wise credit assignment shows significant improvement (Table 3).
3. **Action space:** Search-R1's free-form text queries are harder to optimize than `KG-R1`'s structured actions (4 discrete operations with entity/relation arguments).

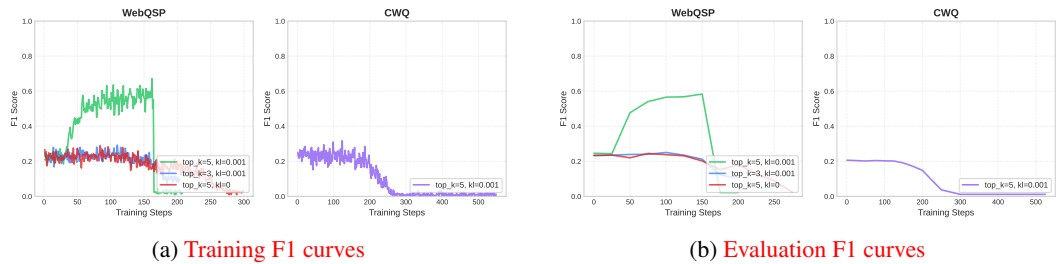

(a) Training F1 curves  (b) Evaluation F1 curves

Figure 10: Training dynamics of Search-R1 on WebQSP (left panel) and CWQ (right panel). All configurations exhibit training instability: `top_k=5, kl=0.001` (green) briefly achieves $\sim$0.58 F1 on WebQSP but collapses after step 150; other configurations remain flat at $\sim$0.22–0.25 F1 before collapsing. On CWQ, F1 starts at $\sim$0.21 and degrades to near zero after $\sim$250 steps.

**Latency Comparison.** We compare the inference efficiency of `KG-R1` and the KG-adapted Search-R1. Table 15 reports latency metrics on WebQSP with batch size 256.

Table 15: Latency comparison between Search-R1 (adapted to KG) and `KG-R1` on WebQSP (batch size 256). **Bold** = best performance.

| Method | Backend | Turns | Retrieval Calls | Gen Tokens | Total Tokens | Time/Q (ms) | QPS |
|---|---|---|---|---|---|---|---|
| Search-R1 | KG embedding server | 3.45 | 2.44 | 298 | 474 | 1115.7 | 3.59 |
| KG-R1 | KG Server | 5.8 | 4.8 | 301 | 3241 | **216.3** | **18.5** |

`KG-R1` is approximately 5$\times$ faster than Search-R1 despite executing more turns (5.8 vs 3.45) and more retrieval calls (4.8 vs 2.44), achieving 5.2$\times$ lower latency (216ms vs 1116ms per question). The critical bottleneck is retrieval latency: Search-R1 relies on embedding-based search, where each query incurs significant overhead ($\sim$100–150ms for embedding computation and cosine similarity calculation). In contrast, `KG-R1`'s graph server performs direct entity/relation lookups in $\sim$2–5ms per call. This demonstrates that structured graph traversal is not only more accurate but also significantly more efficient than embedding-based retrieval approaches.

## H EXAMPLES OF RESPONSES

---

**KG-R1 Response Example — WebQSP**

---

Question: What are Victoria Beckham's kids' names? Initial entities : 'Victoria Beckham'

---

`<think>`
I need to find out the names of Victoria Beckham's kids. I will start by finding any relations to her children.
`</think>`
`<kg-query>`
`get_tail_relations`("Victoria Beckham")
`</kg-query>`

---

`<information>` Tail relations for entity "Victoria Beckham":
base.popstra.celebrity.breakup, award.award_nominee.award_nominations, ..., people.person.children, ... `</information>`

---

`<think>`
The relation `people.person.children` is relevant. I will use it to get the names of her children.
`</think>`
`<kg-query>`
`get_tail_entities`("Victoria Beckham", "people.person.children")
`</kg-query>`

---

`<information>` Tail entities for relation "people.person.children":
Brooklyn Joseph Beckham, Cruz David Beckham, Harper Seven Beckham, Romeo James Beckham `</information>`

---

`<think>`
The names of Victoria Beckham's children are found. I will verify by checking each child's parent relation.
`</think>`
`<kg-query>`
`get_head_relations`("Brooklyn Joseph Beckham")
`</kg-query>`

---

`<information>` Head relations for "Brooklyn Joseph Beckham":
base.godparents.godparent.godchildren, people.person.children, people.sibling_relationship.sibling `</information>`

---

`<think>`
Confirming Brooklyn's parents using `people.person.children`.
`</think>`
`<kg-query>`
`get_head_entities`("Brooklyn Joseph Beckham", "people.person.children")
`</kg-query>`

---

`<information>` Head entities for relation with tail "Brooklyn Joseph Beckham":
David Beckham, Victoria Beckham `</information>`

---

`<think>` Brooklyn, Cruz, Harper, and Romeo all confirm to have David and Victoria Beckham as parents. `</think>`
`<answer>`
Brooklyn Joseph Beckham, Cruz David Beckham, Harper Seven Beckham, Romeo James Beckham
`</answer>`

---

**[ground_truth]** {'Brooklyn Joseph Beckham', 'Romeo James Beckham', 'Cruz David Beckham', 'Harper Seven Beckham'}

---

Figure 11: Example `KG-R1` response on WebQSP, showing multi-step reasoning and verification for a person-children query. Blue denotes responses generated by the `KG-R1` agent.

---

**KG-R1 Response Example - CWQ**

---

Question: What movie, written by Mike Jefferies, starred Nick Cannon?
Initial entities: "Mike Jefferies", "Nick Cannon"

---

```
<think>
```
Let's start by finding a movie written by Mike Jefferies. We can use the relation that associates writers with their movies.
```
</think>
<kg-query>
get_tail_relations("Mike Jefferies") </kg-query>
```

---

`<information>`Tail relations for entity "Mike Jefferies":
film.film_story_contributor.film_story_credits,        people.person.nationality,      people.person.place_of_birth, film.writer.film, ... `</information>`

---

```
<think>
```
From the initial query results, we have two possible options for Mike Jefferies' work. Let's start with "film.writer.film" to find the movies written by Mike Jefferies and then see if Nick Cannon starred in any of those films.
```
</think>
<kg-query>
get_tail_entities("Mike Jefferies", "film.writer.film")
</kg-query>
```

---

`<information>`Tail entities for relation "film.writer.film" with head "Mike Jefferies": Goal II: Living the Dream, Goal III: Taking on the World`</information>`

---

```
<think>
```
Now that we have two movies written by Mike Jefferies, let's check if Nick Cannon has starred in any of these movies. We'll use the relation that connects actors with the movies they star in.
```
</think> <kg-query>
get_head_relations("Nick Cannon")
</kg-query>
```

---

`<information>`Head relations for entity "Nick Cannon": base.popstra.dated.participant, film.personal_film_appearance.person, music.album.primary_release,
base.popstra.lived_with.participant, film.film.written_by, ... `</information>`

---

...
...
...

---

```
<think>
```
From the relations where "Goal II: Living the Dream" is involved, we can see that it is a film with a writer, which aligns with Mike Jefferies.
```
</think>
<answer>
```
Goal II: Living the Dream, School Dance, School Gyrls, Goal III: Taking on the World
```
</answer>
```

---

**[ground_truth]** {'Goal II: Living the Dream' }

---

Figure 12: Example KG-R1 response in CWQ. Blue denotes responses generated by the KG-R1 agent.

---

**KG-R1 Response Example - SimpleQA**

Question: where is the madam satan located? (Initial entities: "Madam Satan")

---

```
<think>
```
I need to find out where Madam Satan is located. First, I will check if it has a location or if it's an entity that can be queried about its location.
```
</think>
<kg-query>
get_tail_relations("Madam Satan")
</kg-query>
```

---

```
<information>
```
Tail relations for entity "Madam Satan":
film.film.cinematography,          common.topic.notable_types,          film.film.language,
film.film.starring,film.film.country,        ...          award.ranked_item.appears_in_ranked_lists,
film.film.film_art_direction_by
```
</information>
```

---

```
<think>
```
None of the identified relations are directly related to location. Let me check if Madam Satan is detailed in a geographical context.
```
</think>
<kg-query>
get_tail_entities("Madam Satan", "film.film.country")
</kg-query>
```

---

```
<information>
```
Tail entities for relation "film.film.country" with head "Madam Satan": United States of America
```
</information>
```

---

```
<think>
```
I have found that Madam Satan, identified as a film, is located in the United States of America. However, it's crucial to confirm the specific location or city within the country where Madam Satan was filmed.
```
</think>
<kg-query>
get_tail_entities("Madam Satan", "film.film.starring")
</kg-query>
```
..
..
..

---

```
<information>
```
Tail entities for relation "film.film.runtime" with head "Madam Satan": m.03kx9f2
```
</information>
```

---

```
<think>
```
I am still not obtaining the most helpful information about the location. Possibly Madam Satan was released, but not filmed in a specific place.
```
</think>
<kg-query>
get_head_relations("m.02tb01n")
</kg-query>
```

---

```
<information>
```
Head relations for entity "m.02tb01n":
film.film.starring, film.actor.film
```
</information>
```

---

```
<think>
```
I have questionned the relation "film.actor.film". This relation includes the movies the actor is in. Let's try to find a clearer detail for the location of Madam Satan.
```
</think>
<answer>
```
United States of America
```
</answer>
```

---

**[ground_truth]** {'United States of America' }

---

Figure 13: Example KG-R1 response in SimpleQA. Blue denotes responses generated by the `KG-R1` agent.

---

**KG-R1 Response Example — T-REx**

Question: What is the instance of Solberg?
 Initial entities: "Solberg"

---

```
<think>
```
Let's start with the entity Solberg and try to find out what it represents. It seems like Solberg could be an entity that describes a geopolitical boundary, location, or characterization. The query can be adjusted from general to specific as needed.
```
</think>
<kg-query>
get_tail_relations("Solberg")
</kg-query>
```

---

```
<information>
```Tail relations for entity "Solberg": P1889, P407, P1552, P282, P31, P460```</information>
```

---

```
<think>
```
The relations provided show how Solberg is related to other entities. Let's explore each relation to find out more about Solberg.
```
</think>
<kg-query>
get_tail_entities("Solberg", "P282")
</kg-query>
```

---

```
<information>
```Tail entities for relation "P282" with head "Solberg": Latin script```</information>
```

---

```
<think>
```
The relation P282 corresponds to "type.object.name" according to the API documentation, indicating that Solberg is a human name. The relation P407 is also noteworthy but does not give further specific information. Let's try a broader query for more context related to Solberg.
```
</think>
<kg-query>
get_tail_entities("Solberg", "P31")
</kg-query>
```

---

```
<information>
```Tail entities for relation "P31" with head "Solberg": Wikimedia disambiguation page, family name```</information>
```

---

...
...
...

---

```
<think>
```
The retrieval of "Soelberg", "Sohlberg", and "Sohlberg" suggests that Solberg might be associated with these names. To find an instance of Solberg, we can search for a specific entity of this name.
```
</think>
<answer>
```
Latin script, Wikimedia disambiguation page, family name, Soelberg, Sohlberg, Sohlberg
```
</answer>
```

---

**[ground_truth]** {'family name', 'human settlement', 'urban area in Sweden', 'minor locality in Sweden', 'minor locality in Sweden', 'Wikimedia disambiguation page', 'human settlement', 'Ortsteil', 'stone setting', 'human settlement', 'minor locality in Sweden', 'hill', 'stone setting', 'hill', 'human settlement' }

Figure 14: Example KG-R1 response in T-REx. Blue denotes responses generated by the `KG-R1` agent.

---

**KG-R1 Response Example - QALD10en**

Question: After whom is the Riemannian geometry named? (Initial entities: "Riemannian geometry")

---

```
<think>
```
I need to find out who the Riemannian geometry is named after. Let's start by investigating the relations and entities connected to "Riemannian geometry".
```
</think>
<kg-query>
get_tail_relations("Riemannian geometry")
</kg-query>
```

---

`<information>`Tail relations for entity "Riemannian geometry":
on focus list of Wikimedia project, named after, maintained by WikiProject, topic's main category, instance of, subclass of`</information>`

---

```
<think>
```
From the relations extracted, it appears the Riemannian geometry is named after someone. I will now search for the person it's named after.
```
</think>
<kg-query>
get_tail_entities("Riemannian geometry", "named after")
</kg-query>
```

---

`<information>`Tail entities for relation "named after" with head "Riemannian geometry": Bernhard Riemann`</information>`

---

```
<think>
```
I now have the information that the Riemannian geometry is named after Bernhard Riemann. Let's confirm this by checking if there's more detailed information about this relationship.
```
</think>
<kg-query>
get_head_relations("Bernhard Riemann")
</kg-query>
```

---

`<information>`Head relations for entity "Bernhard Riemann":
named after`</information>`

---

```
<think>
```
These results are confirming that Bernhard Riemann is indeed the person the Riemannian geometry is named after. No further information is needed.
```
</think>
<kg-query>
get_head_entities("Bernhard Riemann", "named after")
</kg-query>
```

---

`<information>`Head entities for relation "named after" with tail "Bernhard Riemann": Riemannian geometry`</information>`

---

```
<think>
```
This confirms my findings; the Riemannian geometry is named after Bernhard Riemann.
```
</think>
<answer>
```
Bernhard Riemann
```
</answer>
```

---

**[ground_truth]** {'Bernhard Riemann'}

Figure 15: Example KG-R1 response in QALD10en. Blue denotes responses generated by the `KG-R1` agent.

---

**KG-R1 Response Example — GrailQA**

Question: what video game is published by softdisk?
Initial entities: "Near West Side"

---

`<think>`
First, I need to find out what game is published by Softdisk. Without specific knowledge about "Near West Side", I will start with the game publisher for Softdisk.
`</think>`
`<kg-query>`
`get_head_relations`("Near West Side")
`</kg-query>`

---

`<information>`Head relations for entity "Near West Side":
book.periodical_publisher_period.publisher, business.industry.companies, business.employment_tenure.company`</information>

---

`<think>`
The head relation "business.employment_tenure.company" suggests that "Near West Side" might be a game company or related to a game. I will try to find out the publisher of a game related to "Near West Side" by looking for games published by Softdisk.
`</think>`
`<kg-query>`
`get_tail_relations`("Near West Side")
`</kg-query>`

---

`<information>`Tail relations for entity "Near West Side":
organization.organization.place_founded, organization.organization.headquarters, book.periodical_publisher.periodicals_published, business.employer.employees, cvg.cvg_publisher.games_published, business.business_operation.industry, common.topic.notable_types`</information>

---

...
...
...

---

`<think>`
The previous query did not provide a conclusive answer. Let's try a different approach by checking if "Augusto Cury" is associated with any game publishers.
`</think>`
`<answer>`
Augusto Cury, Catacomb 3-D, Chagunitzu, Dangerous Dave in the Haunted Mansion, Hovertank 3D, Iron Seed, Lado Skorin, North Cambridge Academy, Roberto Torretti, Shadow Knights, Star Film Company, Tiles of the Dragon
`</answer>`

---

**[ground_truth]** {'Rescue Rover','Catacomb 3-D','Shadow Knights',"Dangerous Dave's Risky Rescue",'Tiles of the Dragon', 'Catacomb', 'Commander Keen: Keen Dreams', 'Dave Goes Nutz!','Chagunitzu','Dangerous Dave in the Haunted Mansion','Hovertank 3D','Iron Seed'}

Figure 16: Example KG-R1 response in GrailQA. Blue denotes responses generated by the `KG-R1` agent.

---

**KG-R1 Response Example — MultiTQ**

Question: Who negotiated with Malaysia in 2005–10?

---

`<think>`
To answer this, I should check which relations connect to Malaysia that indicate negotiations, then look for counterpart entities within the 2005–10 range.
`</think>`
`<kg-query>`
`get_head_relations`("Malaysia")
`</kg-query>`

---

`<information>` Head relations for entity "Malaysia":
Appeal for aid; Appeal for diplomatic cooperation; Appeal for humanitarian aid; ... Engage in negotiation; Express intent to settle dispute `</information>`

---

`<think>`
"Engage in negotiation" appears among head relations. Next, retrieve the counterpart entities that negotiated with Malaysia.
`</think>`
`<kg-query>`
`get_head_entities`("Malaysia", "Engage in negotiation")
`</kg-query>`

---

`<information>` Head entities connected by `(?, Engage in negotiation, Malaysia)`:
Cambodia [2008-06]; China [2011-11]; Governor (Malaysia) [2007-05]; Iran [2006-05; 2006-07]; Japan [2005-05; 2005-06; 2009-01; 2011-10]; Julia Gillard [2011-07]; Mwai Kibaki [2007-04]; ... Yemen [2005-02; 2008-10] `</information>`

---

`<think>`
We need the counterpart *who* negotiated with Malaysia specifically within 2005–10. Signals suggest Thailand-related negotiations in this window; inspect Thailand's negotiation partners.
`</think>`
`<kg-query>`
`get_tail_relations`("Thailand")
`</kg-query>`

---

`<information>` Tail relations for entity "Thailand":
Accede to demands for rights; Accuse; Appeal for aid; Appeal for de-escalation of military engagement; ... Engage in negotiation `</information>`

---

`<think>`
Confirm the negotiation counterparts of Thailand to identify the specific actor matching the 2005–10 window.
`</think>`
`<kg-query>`
`get_tail_entities`("Thailand", "Engage in negotiation")
`</kg-query>`

---

`<information>` Tail entities connected by `(Thailand, Engage in negotiation, ?)`:
Abhisit Vejjajiva [2009-06; 2009-10; 2010-01]; Cambodia [2005-04; 2005-06; 2008-01; ...; 2012-07]; China [2006-06; 2006-09; 2010-03; 2011-01]; Donald Rumsfeld [2005-06]; Koichiro Matsuura [2011-02]; ... South Korea [2005-12] `</information>`

---

`<think>`
Within 2005–10, the counterpart that negotiated with Malaysia is *Surakiart Sathirathai*.
`</think>`
`<answer>`
Surakiart Sathirathai
`</answer>`

---

**[ground_truth]** {'Surakiart Sathirathai'}

---

Figure 17: Example `KG-R1` response on MultiTQ, illustrating all four query functions and their usage. Blue denotes responses generated by the `KG-R1` agent.

