# OpenReview forum: "Efficient and Transferable Agentic Knowledge Graph RAG via Reinforcement Learning"
_ICLR.cc/2026/Conference — ICLR 2026 Conference Desk Rejected Submission_

### Official Review · Reviewer_mum1 · 2025-10-16

**Soundness:** 4
**Presentation:** 4
**Contribution:** 3
**Rating:** 8
**Confidence:** 3

**Summary:**

This paper proposes KG-R1, an agentic knowledge-graph retrieval-augmented generation (KG-RAG) framework utilizes a single LLM agent with a knowledge graphs (KGs) server through end-to-end reinforcement learning (RL). KG-R1 achieves strong reasoning accuracy with fewer tokens and lower latency and computational cost against baselines. Furthermore, it can transfer to diverse KGs without additional training which shows the ability of extension.

**Strengths:**

1. The design is principled and clear: a single agent interacting with a KG server. The completeness and schema-agnostic design make the transferability inherent from the start.
2. The RL approach with verifiable turn rewards and outcome-level global reward is well-suited for multi-turn KG interaction and backed by detailed equations.
3. The results are promising, showcasing both efficiency and accuracy. The ablation study clearly shows all components are necessary.
4. The plug-and-play features make the framework flexible in deployment across various KGs.

**Weaknesses:**

1. While the experiments provide sufficient validation for the claims of the paper, I believe that the limitations of the method are not properly addressed.
2. There are missing citations in line 918 and 923. Additionally, there are two "of" in line 1514, and two "?" in line 1514 and 1568.

**Questions:**

1. Could you report the limitations and common failure cases of KG-R1?

---

> ### Author Response · Authors · 2025-11-25
>
> Thank you for your thoughtful and encouraging review. We are particularly encouraged by your recognition that our design is "principled and clear," with a schema-agnostic structure that makes transferability inherent. We also appreciate your comment that the RL approach is well-suited for multi-turn KG interaction, and that the results showcase "both efficiency and accuracy."  Below, we respond to each of your comments.
>
> ---
> **1. What are the limitations and common failure modes of KG-R1?**
>
> We list the challenges of KG-R1 and future directions to address these challenges:
>
> - **Restricted Expressivity of 1-hop Retreival Actions**: KG-R1 currently operates on 1-hop relational transitions. While this simplifies the policy and enables a KG-schema agnostic design, it may limit the agent's ability to efficiently retrieve high-level logical patterns or reason over long-range dependencies. Future work can incorporate more elaborate actions, such as multi-hop paths retreival or hierarchical retrieval, to improve retreival efficiecny.
>
> - **Dependency on Entity Linking**: The retrieval actions in KG-R1 currently rely on exact string matching for entity names. Consequently, if the entity matching fails (e.g., the agent generate semantically similar but textually distinct entities), the KG-R1 agent receives an error message and loses the reasoning context. Future work can address this issue by incorporating semantic-based entity linking [1, 2] to allow for fuzzy matching and synonym resolution.
>
> - **Sparse Reward in Multi-Turn RL**: While our proposed multi-turn RL setup is effective, it has large room for advncement, for optimal training and policy leanring. Future work could incorporate early-exit mechanisms [3] to reduce computational redundancy, as well as sequence-level policy optimization [4] to mitigate the high variance inherent in token-level updates, thereby further enhancing RL training stability and final performance.
>
> We will include this detailed discussion in the Limitations section of the updated manuscript.
>
> ---
> **2. Minor typos.**
>
> We appreciate the reviewer’s careful review, which has enabled significant improvements to the paper. We will update the manuscript accordingly.
>
> References
>  - *[1] Wu et al., "Scalable Zero-shot Entity Linking with Dense Entity Retrieval", EMNLP 2020. *
>  - *[2] Huang & Zhao, "Leveraging Large Language Models for Entity Matching", arXiv 2024.*
>  - *[3] Dai et al., "S-GRPO: Early Exit via Reinforcement Learning in Reasoning Models", arXiv 2025.*
>  - *[4] Zheng et al., "Group Sequence Policy Optimization", arXiv 2025.*

---

> > ### Comment · Reviewer_mum1 · 2025-11-27
> > **Thanks for your response**
> >
> > Thank you for the detailed response. The authors have addressed most of my concerns, and I will keep my current rating.

---

### Official Review · Reviewer_wf7t · 2025-10-28

**Soundness:** 3
**Presentation:** 2
**Contribution:** 3
**Rating:** 6
**Confidence:** 2

**Summary:**

This is a interesting paper that tackles a timely and important problem in knowledge graph-augmented generation (KG-RAG). The authors propose KG-R1, a new framework that reframes the KG-RAG task from a complex, multi-stage pipeline into a more streamlined, single-agent system trained with reinforcement learning. The goal is to create a system that is not only efficient but can also be easily transferred to new, unseen knowledge graphs—a major challenge for current methods.

**Strengths:**

The core idea of replacing complex, multi-module pipelines with a single, RL-trained agent that learns to interact with a simple KG "environment" is elegant and well-motivated. The agent learns to perform a series of simple, one-hop retrieval actions over multiple turns, with the entire process optimized end-to-end. This approach directly targets the high inference costs and poor generalizability of many existing KG-RAG systems.

**Weaknesses:**

The "N-run beam search" is a bit of a misnomer. The technique described is essentially running the model independently N times and taking the union of the answers. This is a form of ensembling or self-consistency, not what is typically understood as beam search in the context of sequence generation or graph traversal.

The paper could be more transparent about the practicalities of RL training. Reinforcement learning is known to be difficult to get right. While the training curves in the appendix look clean and stable, it would be valuable to have a more candid discussion about the challenges. How sensitive was the training to hyperparameters? What was the overall engineering effort required to set up the RL pipeline compared to, for example, a standard supervised fine-tuning approach? A brief discussion on this would help readers better understand the trade-offs involved.

**Questions:**

see weaknesses

---

> ### Author Response · Authors · 2025-11-25
>
> Thank you for your very detailed and thoughtful review. We sincerely appreciate your recognition that our core idea is “elegant and well-motivated” and that our approach “directly targets the high inference costs and poor generalizability” of existing KG-RAG systems, as this aligns with our core motivation. We address each of your points in detail below.
>
> ---
> 1. **Confusion around "N-run beam search"**
>
> We appreciate the reviewer’s feedback regarding the term “$N$-beam search”. We agree that “$N$-beam search” implies maintaining a buffer of top-$k$ branches at each turn. To clarify, our implementation executes $N$ independent response samplings in parallel. To eliminate ambiguity, we have renamed the method **$N$-response sampling** throughout the manuscript.
>
> - **What is $N$-response sampling?:** We run model inference $N$ times independently and aggregate the predicted answers. The KG-R1 agent, using a nonzero decoding temperature ($T=0.9$), yields diverse answers in each run due to its stochastic policy. This facilitates a broader search, allowing for the construction of a more complete answer set.
>
> - **Why $N$-response sampling?:** Concretely, in KG-R1, the agent traverses the Knowledge Graph via 1-hop retrieval actions to form reasoning paths. In the current setting with a finite turn horizon, a single run typically yields a single reasoning path $e_1 \xrightarrow{r_1} e_2 \dots \xrightarrow{r_n} e_{\text{final}}$. KGQA benchmarks like WebQSP and CWQ often admit multiple valid reasoning chains and answers; therefore, a single trajectory is often insufficient to predict the full set of answers. Prior works incorporate multiple reasoning paths using various strategies (See the Point 2). To fairly compare with these prior works, we adopt $N$-response sampling.
>
> ---
> 2.  **Sensitivity of training to hyperparameters and comparison to SFT**
> - **Hyperparameter sensitivity.** we found that our training is generally robust to variations in standard hyperparameters such as learning rate and batch size. This stability may benefit from our selection of a KL regularization coefficient ($\beta \in [0.01, 0.02]$) (see Appendix D.4). Although we acknowledge that a comprehensive hyperparameter sweep is necessary to identify the optimal configuration, our primary focus was on establishing the efficacy of the proposed framework. We therefore leave exhaustive parameter tuning for future work.
> - **Comparison with standard supervised fine-tuning (SFT).** Prior works [1,2] indicate that RL with verifiable rewards often robust comparable to SFT. Becuase our training incorporated verifiable reward functions (Section 3.3 and Appendix D.3) using the well establised WebQSP and CWQ datasets, we did not observe significant challenges that destabilized trainig.
>
> ---
> We again thank the reviewer for the many valuable feedbacks, which has helped clarify key aspects of our work. We will revise the manuscript to reflect these points within the next few days.
>
> References
> - *[1] Shao, Zhihong, et al. "DeepSeekMath: Pushing the Limits of Mathematical Reasoning in Open Language Models." arXiv preprint, 2024.*
> - *[2] Shenfeld, Idan, Jyothish Pari, and Pulkit Agrawal. "RL's Razor: Why Online Reinforcement Learning Forgets Less." arXiv preprint, 2025.*

---

> > ### Comment · Reviewer_wf7t · 2025-11-27
> >
> > Thanks for the detailed reply. I will keep my score to support this paper.

---

### Official Review · Reviewer_TvhU · 2025-10-28

**Soundness:** 3
**Presentation:** 3
**Contribution:** 2
**Rating:** 4
**Confidence:** 4

**Summary:**

This paper introduces an agentic KG retrieval-augmented generation (KG-RAG) framework through reinforcement learning (RL), termed KG-R1. To be specific, the proposed method utilizes a single agent to interact with KGs as its environment, which attempts to learn to retrieve at each step and incorporate the retrieved information into reasoning and generation. The overall process is optimized through end-to-end RL, combining turn-wise and outcome-based rewards signals. Extensive experiments show the effectiveness of the proposed method.

**Strengths:**

1.	This paper is well-organized and easy to follow.
2.	Utilizing RL to incorporate LLMs with KGs is interesting.
3.	The proposed method demonstrates both efficiency and transferability.

**Weaknesses:**

1. The integration of retrieval and RL is not novel, as similar approaches have been explored in prior works such as Search-R1 [1], R1-Searcher [2]. The proposed method looks like a straightforward extension of these existing methods to the KG domain, lacking significant innovation.
2. It may lack some representative baseline methods for comparison, such as PoG [3], GCR [4], GNN-RAG [5]. It would be better to include these approaches for a comprehensive comparison.
3. Although the proposed method claims to be efficient, the experiments in Figure 2 seem confusing. The overall token usage of KG-R1 is higher than the baselines, as the proposed method needs to interact with the KG server multiple times and retrieve knowledge for reasoning. It is unfair to only compare the generated tokens without considering the input tokens from the KG server. Both the input and output tokens contribute to the overall cost of using LLMs.
4. For the transferability experiments, it would be more convincing to include results on the MedQA dataset, as the current datasets are primarily derived from Freebase or Wikidata.
5. It would be beneficial to include an analysis of the average number of times the LLMs and the KG server are called for each dataset as well as the total inference time, to provide a clearer picture of the efficiency of the proposed method.

[1] Jin, Bowen, et al. "Search-r1: Training llms to reason and leverage search engines with reinforcement learning." arXiv preprint arXiv:2503.09516 (2025).

[2] Song, Huatong, et al. "R1-searcher: Incentivizing the search capability in llms via reinforcement learning." arXiv preprint arXiv:2503.05592 (2025).

[3] Chen, Liyi, et al. "Plan-on-graph: Self-correcting adaptive planning of large language model on knowledge graphs." Advances in Neural Information Processing Systems 37 (2024): 37665-37691.

[4] Luo, Linhao, et al. "Graph-constrained Reasoning: Faithful Reasoning on Knowledge Graphs with Large Language Models." Forty-second International Conference on Machine Learning.

[5] Mavromatis, Costas, and George Karypis. "Gnn-rag: Graph neural retrieval for efficient large language model reasoning on knowledge graphs." Findings of the Association for Computational Linguistics: ACL 2025. 2025.

**Questions:**

1. Regarding the efficiency experiments shown in Figure 2, please clarify whether the reported token usage refers only to the generated tokens or if it also includes the input tokens. As shown in Appendix F, the token usage appears to be higher.

2. How about the transferability of the proposed method to other datasets, such as MedQA?

3. Could you provide an analysis of the average number of calls made to the LLMs and the KG server for each dataset, as well as the total inference time?

---

> ### Author Response · Authors · 2025-11-25
>
> We thank the reviewer for insightful review and detailed questions. We are sincerely glad that you found the paper *"well-organized and easy to follow."* We also appreciate your highlighting that our approach is *"interesting"* and that our method successfully *"demonstrates both efficiency and transferability"*. Below, we provide our responses to the points you have kindly raised.
>
> ---
> - **1. Novelty and distinction from prior works?**
>
> We acknowledge the integration of GRPO-style RL as explored by Jin, Bowen, et al. (2025) [1] and Ong, Huatong, et al. (2025) [2] which we explicitly cite as motivation in our "Related Work". However, applying this paradigm to structured KGs requires non-trivial innovations beyond a simple extension. We emphasize our three key novelty and distinctions:
>
> - **First RL-fintuned Agent for KGQA:** KG-R1 is the first framework to apply RL to LLM agents within the KG domain. We formulate the KG-RAG reasoning process as a concrete RL problem, effectively bridging the gap between structured graph traversal and LLM reasoning.
> - **Novel Architecture & Framework:** Our framework is designed to address the efficiency and generalization challenges often overlooked in prior works. We propose an architecture of a single agent interacting with a KG retrieval server in multi-turn, reduces the deployment complexity. Furthermore, this design avoids the KG schema-dependency of prior methods, allowing KG-R1 to generalize zero-shot across diverse KGs (Table 2).
> - **Comprehensive Evaluation:** Unlike prior works focused solely on performance, we analyze practical viability as well. We provide a comprehensive analysis that simultaneously addresses token efficiency, reasoning accuracy, and cross-KG transferability across KG-R1 and various baselines (e.g., RoG, ToG, RekNos), reflecting the multifaceted requirements of real-world deployment.
>
> ---
> - **2. Additional baselines results**
>
> We appreciate the suggestion to compare our method against GCR [3], GNN-RAG [4], and PoG [5]. We have conducted these additional experiments and report the results below.
>
> -  **Experimental Setup:** We follow the standard settings of GCR, GNN-RAG, and PoG.
>
>     (1) **Standardization of reasoning LLM:** While the original papers for these baselines typically utilize frontier API models (e.g., GPT-3.5, GPT-4) for their main results, we standardize the reasoning LLM of all baselines to use **Qwen2.5-3B-Instruct**. This ensures a fair comparison with KG-R1 and testing effectiveness of the *methodology* rather than the underlying model strength.
>
>     (2) **GCR Implementation:** Specifically for GCR, we adhered to a fine-tuned LLaMA-3.1-8B model for the path retrieval step, followed by the Qwen2.5-3B-Instruct for the reasoning step.
>
> - **Results:**
>     We present the detailed performance and efficiecny comparison of KG-R1 against baselines (GCR, GNN-RAG, and PoG) in Table 1 below.
>
>     **Table 1: Performance comparison on WebQSP and CWQ.** Bold = best performance.*
>     | Method | WebQSP Hit@1 | WebQSP F1 | WebQSP Total/Gen | CWQ Hit@1 | CWQ F1 | CWQ Total/Gen |
>     | :--- | :---: | :---: | :---: | :---: | :---: | :---: |
>     | GCR (Complete) | 72.3 | 65.5 | **421** / 59 | 56.0 | 46.3 | **453** / 58 |
>     | GNN-RAG | 81.2 | 72.0 | 480 / 31 | 61.7 | 59.4 | 818 / 16 |
>     | PoG | 19.4* | 7.1* | 2,318 / 248 | – | – | 2,292 / 284 |
>     | **KG-R1 (1 run)** | **84.7** | **77.5** | 3,200 / 302 | **73.8** | **70.9** | 3,300 / 377 |
>
> - **Discussion:** Our analysis of these results highlights three key implications:
>
>     (1) Both GCR and GNN-RAG demonstrate high token efficiency (lowest Total/Gen) but achieve lower accuracy compared to KG-R1. This stems from the strict two-step architecture employed by these methods: a specialized retriever (whether a fine-tuned LLM or a GNN) pre-selects the top-K reasoning paths, drastically reducing the input context and generation costs for the downstream reasoning LLM.
>
>     (2) PoG struggles in this setting, showing low F1 scores across tasks. This suggests that the PoG relies heavily on the complex instruction-following capabilities of larger API models (like GPT-4). When constrained to a 3B model, the model fails to generate valid reasoning plans effectively.
>
>     (3) Overall, these experiments demonstrate that **KG-R1 robustly outperforms existing state-of-the-art baselines** when the reasoning model is replaced to a 3B model.

---

> ### Author Response · Authors · 2025-11-25
>
> - **3. The transferability to MedQA**
>
> We appreciate the suggestion to MedQA. We have conducted the additional experiment and report the results below.
>
> - **Experimental Setup:** Unlike other QA, there is no provided KG for MedQA. We choose incorporatng two known medical-related KGs and teted our Kg-r1 on this graphs.
>
>     (1) **DDB KG** : This graph is a processed version of the DDB (Disease Database + DrugBank) [6], adopting the curation piepline in Yasunaga et al. (2021) [7]. It focuses on clinical reasoning between diseases and pharmaceutical treatments.
>
>     (2) **PrimeKG** (Precision Medicine Knowledge Graph): A comprehensive knowledge graph introduced by Chandak et al. (2023) [8]. It integrates high-quality biomedical relationships across diseases, phenotypes, and drugs for precision medicine.
>
>
> - **Results:**
> We present the detailed performance and efficiecny comparison of KG-R1 on MedQA and comparison with the in Table 1 below.
>
>     **Table 2: Performance comparison on MedQA** (Bold = best performance)
>
>     | Method | Accuracy (Avg) | Total / Gen Tokens (Avg) |
>     | :--- | :---: | :---: |
>     | Vanilla (Qwen2.5-3B-it) | 0.41 | 361 / 142 |
>     | CoT (Qwen2.5-3B-it) |  0.27 | 420 / 230 |
>     | KG-R1 (DDB KG) | 0.35 | 1,739 / 284 |
>     | KG-R1 (PrimeKG) | 0.30 | 2,193 / 308 |
>
>
> - **Discussion:** Our analysis of these results highlights two implications:
>
>     (1) **Baseline vs. CoT**: The Vanilla baseline is the most accurate, whereas CoT performs poorly, suggesting that explicit step-by-step reasoning likely does not help for medQA on 3B model.
>
>     (2) **KG-R1 Underperformance**: Both KG-R1 (DDB) and KG-R1 (PrimeKG) performed worse than the Vanilla baseline. Despite the integration of structured knowledge, neither graph variant provided a meaningful accuracy advantage to justify the increased token usage and computational cost compared to the simple Vanilla model.
>
> - **Reasons of KG-R1 Failure**: There are several possible reasons behind this failure.
>
>     (1) **KG Quality Issues**: Both DDB KG and PrimeKG primarily represent surface-level connections between medical entities. Consequently, the specific, causal reasoning paths required to enhance MedQA performance may be missing or insufficient in these KGs.
>
>     (2) **Complexity of MedQA**: Our experiments reveal that MedQA requires both deep domain knowledge and complex reasoning skills, exposing failure modes in two specific dimensions: (1) Retrieval Sparsity: KG-R1’s reliance on finite single-hop queries limits it to retrieving fragmented, sparse retrieval paths rather than a holistic clinical context. (2) Synthesis Failure: The agent struggles to chain these isolated retrieved facts with the detailed question description, failing to generate the accurate answer.
>
>     (3) **Case Study Analysis**: Consider the following MedQA example:
>
>         Question : "A 47-year-old man presents to the emergency department with fever, fatigue, and loss of appetite for 1 week, followed by right shoulder pain, generalized abdominal pain, and paroxysmal cough. He has had diabetes mellitus for 15 years, for which he takes metformin and gliclazide. His vital signs include a temperature of 38.3°C (101.0°F), pulse of 85/min, and blood pressure of 110/70 mm Hg. On examination, he is ill-appearing, and he has a tender liver edge that is palpable approx. 2 cm below the right costal margin. Percussion and movement worsens the pain. Abdominal ultrasonography is shown. Stool is negative for Entamoeba histolytica antigen. Which of the following is the best initial step in management of this patient condition?"
>
>     Over multiple runs, the KG-R1 agent was able to retrieve isolated reasoning paths, such as:
>
>         Run 1: “Diabetes” → phenotype present → “Insulin resistance” → associated with → “Hepatic steatosis”
>         Run 2: “Fever” → parent-child → “Recurrent fever” → phenotype-present → “Caroli disease”
>
>     However, each run yields only a single reasoning path. These individual paths are insufficient to generate the correct holistic answer to the question asking for right treatment (="Antibiotics and drainage"). Also, we found that the 3B base model lacks the medical reasoning capability to connect these retreived into a diagnosis. Since KG-R1 is trained primarily for graph retrieval, it does not offer the advanced medical reasoning skills needed to overcome this model-size constraint.
>
> We identified that the MedQA is worth to be hiliglighted as the current failure mode of KG-R1 and the future direction to improve. We will revise the manuscript to incoporate this point.

---

> ### Author Response · Authors · 2025-11-25
>
> - **4. Justification of efficiency claims : total token usage vs generation tokens.**
>
> We thank the reviewer for this observation. While KG-R1 utilizes more input tokens, our efficiency claim centers on inference latency, which in practice is dominated by generation tokens rather than total tokens.
>
> - **Latency is mainly driven by generation:** In autoregressive LLMs, the main contributor to inference latency is the generation step. Input tokens are processed in a relatively fast parallel prefill step, whereas generation tokens are produced one by one through slower autoregressive decoding and therefore dominate the runtime. For this reason, our efficiency discussion focuses on generation tokens as the most relevant proxy for inference latency.
> - **Why KG-R1 uses more input tokens:** Baselines like RoG, GCR, and GNN-RAG use specialized retrievers to pre-select top-K paths, aggressively reducing input context. In contrast, KG-R1 performs dynamic, sequential retrieval, patching all retrieved entities and relations into the context. This naturally results in higher input usage than static path-wise retrieval methods.
>
> We strongly agree that generation tokens alone are not a complete efficiency measure. For this reason, we report both total tokens (Table 1) and measured latency (Fig. 7a, 7b), providing a transparent and comprehensive comparison.
>
> ---
>
> - **5. Reported token in efficiency experiments shown in Figure 2**
>
> We clarify that the results shown in Figure 2(b) represent the end-to-end generated (output) tokens per question on WebQSP, as noted in the figure legend.
>
> ---
>
> - **6. Total inference time, number of calls to KG server and LLM in Table2**
>
> We provide the table 3 below.
>
> **Table3. Cross-KG Benchmark Efficiency Metrics**
> | Benchmark    | KG Backend | LLM Calls | Server Calls | Gen Tokens | Total Tokens | Time/Q (ms) |
> |--------------|-----------|-----------|--------------|------------|--------------|-------------|
> | WebQSP       | Freebase  | 5.8       | 4.8          | 301        | 3241         | 216.3       |
> | CWQ          | Freebase  | 5.9       | 4.9          | 375        | 3284         | 226.7       |
> | SimpleQA     | Freebase  | 4.2       | 3.2          | 278        | 2839         | 216.3       |
> | GrailQA      | Freebase  | 5.7       | 4.7          | 408        | 3579         | 226.7       |
> | T-REx        | Wikidata  | 4.1       | 3.1          | 270        | 2882         | 216.3       |
> | QALD-10-en   | Wikidata  | 4.4       | 3.4          | 278        | 3068         | 216.3       |
> | Zero-Shot RE | Wikidata  | 5.7       | 4.7          | 361        | 3445         | 226.7       |
> | MultiTQ      | Temporal  | 5.7       | 4.7          | 434        | 3578         | 226.7       |
>
> - With the agent trained using a maximum of $H=5$ reasoning turns, we observe that the actual number of LLM calls per question ranges from ~4 to 6 across benchmarks.
> - Each turn consists of one LLM call followed by one KG server call, and the final turn LLM call is used to generate the answer, the number of server calls is always exactly one less than the number of LLM calls.
> - Both generated tokens and total tokens then scale almost linearly with the number of KG server calls, while the per-question throughput remains nearly constant (≈210–230 ms) across all datasets.
> ---
> We again thank the reviewer for the many valuable feedbacks, which has helped to improve our work. We will revise the manuscript to reflect these points within the next few days.
>
> References
>
> - *[1] Jin, Bowen, et al. "Search-r1: Training llms to reason and leverage search engines with reinforcement learning." arXiv preprint, 2025.*
> - *[2] Song, Huatong, et al. "R1-searcher: Incentivizing the search capability in llms via reinforcement learning." arXiv preprint, 2025*
> - *[3] Luo, L., et al. Graph-constrained Reasoning: Faithful Reasoning on Knowledge Graphs with Large Language Models. ICML 2025*
> - *[4] Mavromatis, C., & Karypis, G. GNN-RAG: Graph Neural Retrieval for Large Language Model Reasoning. 2024.*
> - *[5] Tan, X., et al. (2025). Paths-over-Graph: Knowledge Graph Empowered Large Language Model Reasoning. In Proceedings of the ACM Web Conference 2025 (WWW ’25), pp. 3505–3522.*
> - *[6] Wishart, D. S., et al. (2018). DrugBank 5.0: a major update to the DrugBank database for 2018. Nucleic Acids Research, 46(D1), D1074-D1082.*
> - *[7] Yasunaga, M., et al. (2021). QA-GNN: Reasoning with Language Models and Knowledge Graphs for Question Answering. Proceedings of NAACL.*
> - *[8] Chandak, P., et al. (2023). Building a knowledge graph to enable precision medicine. Scientific Data, 10(1), 67.*

---

> > ### Comment · Reviewer_TvhU · 2025-11-26
> >
> > Thank you for the detailed response. I have further questions regarding baselines and efficiency:
> >
> > 1. Given the similarity between the proposed method and previous RL-based RAG methods, it would be valuable to compare performance with Search-r1 and R1-searcher on KGQA tasks. This comparison is crucial to demonstrate that adapting to KG scenarios cannot be solved by prior methods. One simple approach could be using the embedding model to encode KG triples and applying Search-r1 and R1-searcher for retrieval and KGQA
> >
> > 2. The paper currently compares only latency and throughput (Tables 3 and 9) with vanilla baselines and CoT. It would be more informative to compare with multi-step methods like Search-r1, R1-searcher, and ToG on QA performance, inference time, total tokens, and tokens used per second, to better illustrate efficiency and effectiveness.
> >
> > 3. Regarding the $N$-beam search concern raised by another reviewer, I am curious how token consumption and inference time are calculated. Are they summed over multiple generated responses, or measured per single response? Would the usage $N$-beam search introduce additional latency and token cost?
> >
> > I will consider changing my score if these questions have been properly answered.

---

> > > ### Author Response · Authors · 2025-11-27
> > >
> > > Thank you for raising these additional insightful comments. We provide our initial responses below.
> > >
> > > - **Comparison with Search-R1 and R1-Searcher:** We value your suggestion that applying R1 style training in a KG-specific scenario and reporting both performance and efficiency would be important for demonstrating our novelty. The idea of creating embeddings for KG triples also sounds like a good initial starting point. We plan to run this experiment and provide the results soon.
> > > - **Token calculation in $N$-beam search:** As we described in the rebuttal for 2HG5, in our method, $N$-response sampling involves $N$ independent inference runs in parallel. In practice, we run $N=8$ response samplings per question by copying the same prompt and executing them with vLLM batching, and we compute accuracy based on 3 bootstrapped responses drawn from these 8 runs. Both the total token count and the generated-token statistics reported for the 3-run setting are calculated as three times the usage of a single run. Because these runs are batched in parallel, this does not increase latency.
> > >
> > > We sincerely appreciate your time and thoughtful questions, and we will provide additional results.

---

> > > > ### Comment · Reviewer_TvhU · 2025-11-27
> > > >
> > > > > In practice, we run $N=8$ response samplings per question by copying the same prompt and executing them with vLLM batching, and we compute accuracy based on 3 bootstrapped responses drawn from these 8 runs.
> > > >
> > > > While awaiting the comparison results, I have two questions about this implementation. First, what criteria should we use to select three responses from the eight runs? Second, if the experiment involved eight runs to obtain the final answers (even though only three are ultimately selected), should we sum the token cost of all eight runs?

---

> ### Author Response · Authors · 2025-12-02
> **Response to Reviewer TvhU’s follow-up questions (1)**
>
> Thank you for raising follow-up questions. We provide responses below.
>
> ---
>
> **Follow-up Q1: Comparison with Search-R1 on KGQA Tasks**
>
> We thank the reviewer for this valuable suggestion. We have conducted the requested comparison experiment, adapting Search-R1 [1] to the KGQA setting.
>
>
> - **Experimental Setup**
>
>      **Adopting KG for Search-R1's embedding-based search:** : We converted KG triples to text format, serializing each triple `(head, relation, tail)` as `"head | relation | tail"` for embedding. We used **BGE-base-en-v1.5** [2] as the embedding model, following Search-R1's KG retrieval server configuration. The entire Freebase subgraph used by WebQSP/CWQ (~1.8M triples) was indexed using FAISS flat index with L2 distance for exact nearest-neighbor search.
>
>     **Search Configuration:** : Following Search-R1 [1]'s original configuration, the LLM agent makes retrieval actions through `<search>` queries in multiple turns (N=5). Retrieved triples (top-k values of 3 and 5, based on cosine similarity) are inserted into `<information>...</information>` tags in Search-R1 format:
>
>     ```
>     <search>Nicolas Cage spouse</search>
>
>     <information>Nicolas Cage --[place of birth]--> Long Beach; Coppola family --[members]--> Nicolas Cage; Nicolas Cage --[nationality]--> United States of America; Nicolas Cage --[is reviewed]--> Place of birth; Nicolas Cage --[is reviewed]--> Country of nationality</information>
>     ```
>
>     **Model Configuration:** We used Qwen2.5-3B-Instruct as the base LLM (same as KG-R1 for fair comparison). Following Search-R1's best configuration, we used PPO, where `<information>` tokens are excluded from the policy gradient loss (gradients do not flow through retrieved content; which termed "state_masking" in Search-R1). For WebQSP, we tested three training configurations: (1) `top_k=5, kl=0` (no KL penalty), (2) `top_k=3, kl=0.001`, and (3) `top_k=5, kl=0.001`. For CWQ, we used `top_k=5, kl=0.001`.
>
>
> - **Results**
>
>     Table. Main Comparison Table between Search-R1 and KG-R1
>
>     | Method | Config | WebQSP F1 | CWQ F1 |
>     | :--- | :--- | :--- | :--- |
>     | Search-R1 | top_k=5, kl=0 | 22.0* | -- |
>     | Search-R1 | top_k=3, kl=0.001 | 25.0* | -- |
>     | Search-R1 | top_k=5, kl=0.001 | 58.0* | 21.0* |
>     | **KG-R1 (Ours)** | -- | **77.5** | **70.9** |
>
>     *Search-R1 results report **peak performance** during training. Due to training instability, final performance collapses to near-zero (see Appendix, Section "Comparison with Search-R1"). KG-R1 results are stable final performance.
>
>
> - **Discussions**
>
>     **Search-R1 fails to learn stable KGQA policies.** Despite extensive hyperparameter tuning, Search-R1 cannot maintain stable performance on KGQA. Even when Search-R1 briefly achieves 58 F1, it collapses to near-zero within ~50 training steps.
>
>     **Why does Search-R1 fail?** While the fundamental cause requires further investigation, we conjecture three contributing factors: (1) **Retrieval noise**: Top-k embedding retrieval returns semantically similar but often irrelevant triples, introducing noisy context that destabilizes policy learning. (2) **Credit assignment**: Search-R1 uses trajectory-level advantage estimation, whereas KG-R1's turn-wise credit assignment shows significant improvement (Table 3, main text). (3) **Action space**: Search-R1's free-form text queries are harder to optimize than KG-R1's structured actions (4 discrete operations with entity/relation arguments).
>
> Training curves and evaluation results are included in the updated manuscript (Appendix, Section G. "Comparison with Search-R1").

---

> ### Author Response · Authors · 2025-12-02
> **Response to Reviewer TvhU’s follow-up questions (2)**
>
> - **Follow-up Q2: Efficiency Comparison with Search-R1**
>
> We compare the inference efficiency of KG-R1 and the KG-adapted Search-R1 described in Q1. Other methods such as ToG and ReKnoS rely heavily on proprietary API models (GPT-3.5, GPT-4o-mini), making direct latency comparison challenging due to variable API response times and costs.
>
>
> Table. Latency Comparison Table (WebQSP, Batch Size 256)
>
> | Method | Backend | Turns | Retrieval Calls | Gen Tokens | Total Tokens | Time/Q (ms) | Q per second |
> |--------|---------|-------|-----------------|------------|--------------|-------------|-----|
> | Search-R1 (adapted to KG) | KG embedding server (Freebase)  | 3.45 | 2.44 | 298 | 474 | 1115.7 | 3.59 |
> | KG-R1 | KG Server (Freebase) | 5.8 | 4.8 | 301 | 3241 | **216.3** | **18.5** |
>
> *Bold = best performance.
>
> - Discussion
>
>     **KG-R1 is ~5x faster than Search-R1** despite executing more turns (5.8 vs 3.45) and more retrieval calls (4.8 vs 2.44), yet achieves 5.2x lower latency (216ms vs 1116ms per question). The critical bottleneck is retrieval latency. Search-R1 relies on embedding-based search, where each query incurs significant overhead (~100-150ms for embedding computation and cosine similarity calculation). In contrast, KG-R1's graph server performs direct entity/relation lookups in ~2-5ms per call. This demonstrates that KG-R1's structured graph traversal is not only more accurate but also significantly more efficient than embedding-based retrieval approaches.
>
> ---
>
> - **Follow-up Q3: N-Response Sampling Methodology**
>
> For the N=3 results reported in Table 1, we generated 8 independent responses per question and then performed bootstrap sampling: we randomly selected 3 responses from the 8 available runs and applied answer set merging to obtain the final answer. This process was repeated 20 times, and we report the average F1 and exact match (Hit@1) scores across these 20 bootstrap iterations. This bootstrap methodology is solely for obtaining precise evaluation metrics with reduced variance; in practical inference, one simply generates N responses directly without bootstrapping.
>
> **Practical Efficiency of N-Response Sampling:** Since each response is generated independently, all N responses can run fully in parallel. This yields O(1) time complexity and O(N) memory complexity with respect to N, making N-response sampling practical for deployment.
>
>
> ---
>
>
> **References**
> [1] *Jin et al., "Search-R1: Training LLMs to Reason and Leverage Search Engines with Reinforcement Learning", arXiv:2503.09516, 2025.*
> [2] *Xiao et al., "C-Pack: Packed Resources For General Chinese Embeddings", arXiv:2309.07597, 2023.*

---

### Official Review · Reviewer_2HG5 · 2025-11-02

**Soundness:** 2
**Presentation:** 3
**Contribution:** 3
**Rating:** 4
**Confidence:** 4

**Summary:**

This paper introduces KG-R1, a RL agentic fine-tuning methodology for knowledge graph guided question answering. It designs a KG agnostic action space and adopts the turn-reward to overcome the limitation of sparse reward.

**Strengths:**

1. The experiments and training details are comprehensive. I quite like the ablation study of the paper, includign the reward hacking of PPO and response example to help validate the quality of RL fine-tuning.

2. The performance improvments are significant, especially compared with recent baselins like ToG and ToG 2.0 on in-domain settings.

**Weaknesses:**

1. The argument of cost saving and zero-shot generalization capability doesn't seem very convincing to me. First of all, the proposed method uses N-beam search and it's not clear whether the baselines use the same strategy and the token efficiency seems to be 1-beam search's token count.

2. What's the performance of ToG2.0 in Table 2? Is there any reason it's missing from the table?

**Questions:**

1. In section 4.1, the author mentioned the N-run beam search, I am wondering whether it's fair for prompt-based baselines, since it includes N-run's total result. What is the value of N in practice and whether it has been calculated into Figure 2 and Table 1's token efficiency.

**Details Of Ethics Concerns:**

N.A.

---

> ### Author Response · Authors · 2025-11-25
>
> We thank the reviewer for their insightful feedback and for accurately summarizing KG-R1’s design of a "KG agnostic action space." We are glad that you found our "experiments and training details are comprehensive" and appreciated the ablation study, specifically our analysis of the reward hacking of PPO. We also value your recognition that our "performance improvements are significant, especially compared with recent baselines like ToG and ToG 2.0."  We address each of your points in detail below.
>
> ---
> **1. What is $N$-beam search?**
>
> We appreciate the reviewer’s feedback regarding the term “$N$-beam search”. We agree that “$N$-beam search” implies maintaining a buffer of top-$k$ branches at each turn. To clarify, our implementation executes $N$ independent response samplings in parallel. To eliminate ambiguity, we have renamed the method **$N$-response sampling** throughout the manuscript.
>
> - **What is $N$-response sampling?:** We run model inference $N$ times independently and aggregate the predicted answers. The KG-R1 agent, using a nonzero decoding temperature ($T=0.9$), yields diverse answers in each run due to its stochastic policy. This facilitates a broader search, allowing for the construction of a more complete answer set.
>
> - **Why $N$-response sampling?:** Concretely, in KG-R1, the agent traverses the Knowledge Graph via 1-hop retrieval actions to form reasoning paths. In the current setting with a finite turn horizon, a single run typically yields a single reasoning path $e_1 \xrightarrow{r_1} e_2 \dots \xrightarrow{r_n} e_{\text{final}}$. KGQA benchmarks like WebQSP and CWQ often admit multiple valid reasoning chains and answers; therefore, a single trajectory is often insufficient to predict the full set of answers. Prior works incorporate multiple reasoning paths using various strategies (See the Point 2). To fairly compare with these prior works, we adopt **$N$-response sampling**.
>
> ---
> **2. How other baselines incorporate multiple reasoning paths and what is the fair comparison?**
>
> Baselines incorporate multiple reasoning paths via two primary strategies:
>
> **Table 1: Comparison of multiple reasoning path strategies in baselines.**
> | Category | Baselines | Mechanism |
> | :--- | :--- | :--- |
> | Step-wise Beam Search | ToG, RekNos, GCR | Maintains top-$N$ partial paths at each hop, pruning low-prob branches to find the single most likely KG reasoning path. |
> | Top-K Pre-selection | RoG, GNN-RAG | Identifies the top-$K$ relation paths by importance and retrieves. |
> | $N$-Response Sampling | KG-R1 (Ours) | Executes $N$ independent stochastic rollouts (full episodes) to maximize exploration. |
>
> **Control of Reasoning Paths ($N$)**: Prior methods typically utilize N=1 to 5 reasoning paths (via beam width or top-k retrieval) to generate the final answer. To ensure a fair comparison, we set $N=3$ for our $N$-response sampling which falls in this same range. This control ensures that our performance improvements are attributed to the effectiveness of independent sampling, rather than simply utilizing a higher number of candidate paths than previous works.
>
> ---
> **3. How is token efficiency of $N=3$ response sampling measured?**
>
> In our methods, $N$-response sampling involves $N$ independent inference runs in parallel. Consequently, the token usage statistics reported for the $N=3$-run configuration in Table 1 and Table 2 are calculated as three times the usage of a single run. This applies to both the total token count and the generated token count, as we assume no shared computation or prompt caching benefits in this specific metric.
>
> ---
> **4. Missing ToG2.0 in Table 2?**
>
> Thank you for raising this concern. We did not initially include ToG 2.0 [1] into this experiment because this method requires a document associated with each entry in the KG. Therefore, this method is only available for Wikipedia-based KG datasets. However, in our cross-KG experiments, we use KGs from different resources, where the documents may not be available. Nevertheless, the method is applicable to the two datasets in our cross-KG experiments: T-REx and QALD-10en, as they use a Wikipedia-based KG. Their performance is 80.7/77.5, and 51.9/54.6 for these two datasets, respectively, in terms of F1 / Hit@1.
>
> ---
> References
> - [1] Sun et al. "Think-on-graph: Deep and responsible reasoning of large language model on knowledge graph", ICLR, 2024

---

> > ### Comment · Reviewer_2HG5 · 2025-11-26
> >
> > Thanks for the response, it addresses most of my concerns and I raised my score. However, I agree with the other reviewer that the novelty of the R1-* like training is limited.

---

> > > ### Author Response · Authors · 2025-11-27
> > >
> > > We are glad to hear that our rebuttal has addressed most of your concerns. Thank you again for your careful review and for increasing your score. We will try to provide an additional experiment on R1-* training in the KG scenario to demonstrate our work's novelty, as Reviewer TvhU suggested. We sincerely appreciate your time and thoughtful comments.

---

### Author Response · Authors · 2025-12-02
**Revision note**

Dear PCs, SACs, ACs, and Reviewers,

Thank you for your valuable suggestions. **To address the comments raised by the reviewers, we have made the following adjustments to the paper**. Changes are highlighted in red.

- We renamed "N-run beam search" to "N-response sampling" throughout the manuscript to eliminate ambiguity, and clarified the methodology: we generate 8 independent responses per question and perform bootstrap sampling (randomly selecting 3 responses and applying majority voting), repeated 20 times. *(Reviewers 2HG5, wf7t, TvhU)*

- We added three additional LLM+KG baselines to Table 1 (Performance and Efficiency Comparison): GCR, GNN-RAG, and PoG, all using Qwen2.5-3B-Instruct for fair comparison with KG-R1. *(Reviewer TvhU)*

- We added ToG 2.0 results to Table 2 (Cross-KG Transferability) for T-REx and QALD-10-en datasets. We note that ToG 2.0 requires a Wikipedia-based KG with associated documents, hence only applicable to these two datasets. *(Reviewer 2HG5)*

- We added Search-R1 adapted to the KGQA setting as a new baseline. KG-R1 significantly outperforms Search-R1 (77.5 vs 58.0* F1 on WebQSP, 70.9 vs 21.0* F1 on CWQ), and is ~5× faster (216ms vs 1116ms per question). Full experimental details and training curves are provided in the Appendix. *(Reviewers TvhU, 2HG5)*

- We discussed the clear distinction from RL-based web search methods (Search-R1, R1-Searcher) in Related Work. *(Reviewer TvhU)*

- We added a new table in the Appendix reporting detailed cross-KG efficiency metrics for KG-R1, including LLM calls, server calls, generated tokens, total tokens, and inference time per question across all benchmarks. *(Reviewer TvhU)*

- We added a new Limitations and Future Work subsection to the Conclusion identifying three key limitations: (1) restricted expressivity of 1-hop retrieval actions, (2) dependency on entity linking with exact string matching, and (3) sparse reward in multi-turn RL. We discuss potential solutions including multi-hop path retrieval, semantic-based entity linking, and sequence-level policy optimization. *(Reviewer mum1)*

- We added MedQA as a failure mode analysis where KG-R1 underperforms vanilla baselines (25.6% vs 31.6% accuracy with DDB KG). This highlights that KG-RAG benefits depend critically on KG quality and domain alignment. Full analysis with two medical KGs is provided in the Appendix. *(Reviewer TvhU)*

We sincerely appreciate the time and consideration you devoted to reviewing our manuscript. Your thoughtful feedback has substantially strengthened the paper.

Sincerely,
The Authors

---

### Note · Program_Chairs · 2026-01-17
**Submission Desk Rejected by Program Chairs**

The following references in this submission do not refer to real documents and/or have major errors in bibliographic information:

 Gautier Izacard, Patrick Lewis, Maria Lomeli, Lucas Hosseini, Fabio Petroni, Stanislaw Jastrzebski, Sebastian Riedel, and Edouard Grave. Atlas: Reasoning over language models with retrieval. Journal of Machine Learning Research, 24, 2023. URL https://jmlr.org/papers/ v24/22-1381.html.
Hao Xiong, Zifeng Wang, Zhijian-huang, Hao tian jia, Yefan-huang, Cheng zhong xu, Zheng-li, Zhi hong chen, Zhi yuan liu, and Zhong zhen su. Knowledge-augmented language model for clinical question answering, 2023.